# Diet and Lifestyle Interventions in Metabolic Dysfunction-Associated Fatty Liver Disease: A Comprehensive Review

**DOI:** 10.3390/ijms26199625

**Published:** 2025-10-02

**Authors:** Muhammad Y. Sheikh, Muhammad F. Younus, Annie Shergill, Muhammad N. Hasan

**Affiliations:** Fresno Clinical Research Center, Fresno, CA 93720, USA; faraz.younus@hotmail.com (M.F.Y.); shergill7590@yahoo.com (A.S.); nameer.hasan@gmail.com (M.N.H.)

**Keywords:** metabolic dysfunction-associated steatotic liver disease (MASLD), metabolic dysfunction-associated steatohepatitis (MASH), macronutrients, micronutrients, lifestyle intervention, weight loss, obesity, dietary modifications, aerobic exercise, resistance exercise

## Abstract

Metabolic dysfunction-associated steatotic liver disease (MASLD) and its progressive form, metabolic dysfunction-associated steatohepatitis (MASH), have become the leading causes of chronic liver disease worldwide, with increasing rates of cirrhosis, hepatocellular carcinoma, and cardiovascular complications. Pathogenesis involves a complex interplay of dietary excess, sedentary lifestyle, insulin resistance, adipose tissue dysfunction, and alterations in the gut microbiome, which collectively lead to hepatocellular stress, inflammation, and fibrogenesis. Despite ongoing advances in pharmacotherapy, lifestyle intervention remains the cornerstone of management. Evidence shows that sustained weight loss of ≥5% reduces hepatic steatosis, ≥7% improves necroinflammation, and ≥10% stabilizes or reverses fibrosis. Dietary strategies, including Mediterranean-style patterns, high-protein approaches, and intermittent fasting, have been shown to be effective in improving insulin sensitivity and reducing intrahepatic triglycerides. Exercise interventions, focusing on both aerobic fitness and resistance training, enhance metabolic flexibility and combat sarcopenia, thereby improving hepatic and systemic outcomes. Equally important are behavioral support, digital health tools, and multidisciplinary approaches that enhance adherence and address barriers such as socioeconomic disparities, limited access, and patient engagement issues. Personalized nutrition plans, integrating physical activity, and ongoing support for behavioral change are essential for long-term disease management. This review synthesizes current evidence on the roles of macronutrients, micronutrients, dietary quality, physical activity, and adjunctive behavioral strategies in managing MASLD. By translating mechanistic insights into practical, evidence-based recommendations, we aim to provide clinicians, dietitians, and exercise professionals with effective frameworks to slow disease progression and improve outcomes across diverse patient populations.

## 1. Introduction

Metabolic dysfunction-associated steatotic liver disease (MASLD) has become the most common chronic liver disease worldwide, now recognized as a multisystem metabolic disorder closely linked to obesity, type 2 diabetes mellitus (T2DM), and cardiovascular risk [1]. A meta-analysis of 72 studies from 17 countries estimated a global prevalence of 32.4%, with a significant increase over the past two decades [2]. Another pooled analysis found that the global prevalence of MASLD rose from 25.3% in 1990–2006 to 38.0% in 2016–2019 [3], highlighting its rapid growth. These rates exceed 50% in high-risk populations. Overall, MASLD prevalence worldwide ranges from 25% to 38%, with future projections suggesting continued growth in the coming decades.

MASLD includes a range of histological changes, from benign isolated steatosis to metabolic dysfunction-associated steatohepatitis (MASH), which is marked by steatosis, lobular inflammation, and hepatocyte ballooning degeneration [4]. Epidemiologic models project that MASH prevalence will increase to 2.8–4.6% of adults by 2040. Transition rates show the disease’s progressive nature: about 6–7% of individuals with simple steatosis develop early fibrosis each year, and 5–9% progress annually to more advanced fibrosis stages without treatment. Approximately 20% of MASLD patients progress to MASH, with 15–20% of those with MASH potentially developing cirrhosis over 15–20 years [5].

Insulin resistance (IR) contributes to the development of MASLD, often alongside obesity, type 2 diabetes, and other aspects of metabolic syndrome [1]. However, many patients have non-obese or lean MASLD, where unchangeable genetic predispositions and modifiable epigenetic and lifestyle factors such as gut dysbiosis, poor diets, and physical inactivity play key roles. Polymorphisms in different loci (e.g., PNPLA3, TM6SF2) increase susceptibility to steatosis, inflammation, and fibrosis [1,6,7]. The risk of adverse liver outcomes is higher with T2DM or cardiovascular disease, with cirrhotic MASH patients facing an annual decompensation risk of over 2.7% and a hepatocellular carcinoma (HCC) risk of 0.1–0.2% per year. Besides liver-related issues, MASLD is increasingly recognized as a factor in extrahepatic diseases, including cardiovascular, kidney, and cancer-related complications, which often surpass liver-related mortality.

Since MASLD is a key part of the cardio-metabolic disease spectrum, the American Diabetes Association (2025) guidelines recommend routine MASLD screening in people with diabetes, along with early lifestyle therapy to reduce hepatic and cardiovascular risks [8]. Terminology changes from NAFLD to MAFLD were introduced to highlight the central role of metabolic dysfunction and to clarify phenotypes [9]. These updates build on earlier efforts to refine disease drivers and heterogeneity in at-risk populations [10]. T2DM, in particular, accelerates fibrosis progression, emphasizing the importance of careful MASLD monitoring in this group [11]. Notably, advanced fibrosis remains the strongest predictor of liver-related and all-cause mortality, underscoring the need for early risk stratification [12]. This connection to cardio-metabolic comorbidities is evident in their prevalence within MASLD, with obesity seen in up to 82% of cases, metabolic syndrome in 66–76%, and T2DM in 55–70%. The risk of cardiovascular disease nearly doubles in MASLD, with 37–73% of affected individuals having established CVD or risk factors. Hypertension is very common, especially in advanced disease, while renal dysfunction and sleep apnea often coexist and increase morbidity. MASLD accounts for 20–40% of cryptogenic cirrhosis, and population studies estimate cirrhosis prevalence of up to 1.8%.

Most patients with MASLD are asymptomatic. However, some nonspecific symptoms may include fatigue, discomfort in the upper right abdomen, abdominal swelling, and pain. The disease is usually brought to medical attention when a patient shows abnormal liver chemistry tests and features of metabolic syndrome. Liver biopsy remains the gold standard for diagnosing MASH, evaluating the severity of steatosis, identifying liver inflammation, hepatocyte ballooning, and fibrosis. However, liver biopsy has limitations due to the risk of complications, its invasive nature, high costs, sampling variability, and intra- and inter-observer differences [13]. As a result, non-invasive tests (NITs) are crucial for clinical assessment. NITs include biochemical biomarkers such as the NAFLD fibrosis score (NFS), Fibrosis-4 (FIB-4) index, Enhanced Liver Fibrosis (ELF) test, and, more recently, the metabolomics-advanced steatohepatitis fibrosis score (MASEF), which provide valuable risk stratification tools. FIB-4 is recommended as a first-line point-of-care test, with secondary assessment advised for patients with an FIB-4 score of ≥1.3. Imaging-based NITs include vibration-controlled transient elastography (VCTE), magnetic resonance elastography (MRE), MR-MASH score, and multiparametric MRI, such as corrected T1 (cT1). These tests help evaluate the likelihood of advanced disease. Combining or sequentially applying these tools improves their accuracy and positive predictive value, thus reducing the number of patients with indeterminate results [14,15,16,17,18]. Recently, Kanwal et al. outlined the clinical care pathway for risk stratification and management for physicians treating patients with MASLD.

Pharmacologic management remains an active area of research. In 2024, Resmetirom, an oral, liver-targeted, thyroid hormone receptor beta (THR-β) selective agonist, received FDA approval under the accelerated program after phase 3 RCT results demonstrated MASH resolution, with a reduction in MASLD activity score of ≥2 points and a decrease in fibrosis by ≥1 stage [19]. Glucagon-like Peptide-1 Receptor Agonists (GLP-1 RAs), especially Semaglutide, have shown particular effectiveness in reducing steatosis and improving liver fibrosis, resulting in MASH resolution after 72 weeks of treatment in moderate-to-advanced MASH [20].

Since a nutritionally imbalanced diet and sedentary lifestyle are the main contributors to the development of MASLD [21], lifestyle modification remains the core approach to management. Targeted changes in diet and physical activity can influence the natural course of the disease, decrease hepatic steatosis, and slow fibrosis progression. Therefore, this comprehensive review explores the role of lifestyle interventions, including dietary strategies and exercise methods, in managing MASLD, with an emphasis on mechanistic pathways, treatment effectiveness, and future directions.

## 2. Pathogenesis of MASLD

MASLD pathogenesis is multifactorial, involving various genetic, metabolic, immunologic, and environmental factors that lead to common downstream processes such as lipotoxicity, oxidative stress, mitochondrial dysfunction, hepatocellular injury, and fibrosis. For clarity and simplicity, the main categories of contributing factors can be outlined as (i) genetic and epigenetic susceptibility, (ii) hepatic lipid metabolism dysfunction, (iii) insulin resistance and systemic metabolic issues, (iv) hormonal and endocrine influences, (v) cardio-metabolic risk factors, (vi) immune and inflammatory pathways, (vii) gut-liver axis and microbial metabolites, and (viii) environmental, socioeconomic, and lifestyle factors.

Genetic factors greatly influence the risk and progression of MASLD. Single-nucleotide polymorphisms in genes such as Patatin-like phospholipase domain-containing protein 3 (PNPLA3), Transmembrane 6 Superfamily Member 2 (TM6SF2), Membrane-bound O-acyltransferase domain-containing 7 (MBOAT7), 17-β-hydroxysteroid dehydrogenase 13 (HSD17B13), and glucokinase regulator (GCKR) affect lipid regulation, triglyceride export, and inflammatory responses. For example, variants in PNPLA3 are associated with lipid accumulation and inflammation in the liver, which can lead to fibrogenesis [22]; variants in TM6SF2 reduce very-low-density lipoprotein (VLDL) export and promote hepatic steatosis [23]; and loss-of-function mutations in HSD17B13 offer some protection against steatohepatitis and fibrosis [23]. In addition to single-gene variants, epigenetic changes—including DNA methylation, chromatin remodeling, and non-coding RNAs like microRNAs—are increasingly recognized as factors influencing MASLD progression, although their exact roles still need further investigation.

Hepatic steatosis occurs due to the dysregulation of multiple lipid-processing pathways. Excessive fatty acid uptake, increased de novo lipogenesis (especially with high fructose consumption), and decreased fatty acid oxidation successively cause the buildup of triglycerides, free cholesterol, and lipotoxic lipid intermediates. This imbalance induces metabolic stress within the hepatocyte and makes the liver more prone to lipotoxic injury [24].

Systemic insulin resistance leads to increased lipolysis in adipose tissue and higher levels of free fatty acids, which then flow to the liver. Insulin resistance in hepatocytes diminishes the suppression of gluconeogenesis but still encourages lipogenesis, which worsens steatosis and lipotoxicity. The accumulation of diacylglycerols, ceramides, and saturated fatty acids triggers oxidative stress, endoplasmic reticulum stress, and mitochondrial dysfunction, resulting in hepatocyte apoptosis and fibrogenesis [25].

Endocrine factors heavily influence MASLD susceptibility and progression. Thyroid hormone deficiency worsens lipid accumulation and fibrosis, while lower levels of growth hormone and adiponectin contribute to insulin resistance and inflammation. Other hormonal regulators, including sex hormones and glucocorticoids, also impact liver fat metabolism and overall metabolic health [26].

MASLD exists within a broader context of cardio-metabolic conditions. Type 2 diabetes, visceral obesity, dyslipidemia, and high blood pressure collectively accelerate hepatic steatosis, inflammation, and fibrogenesis. These interconnected risk factors increase both liver disease severity and cardiovascular disease risk, emphasizing MASLD as both a liver and extra-hepatic condition.

Hepatocellular injury triggers both innate and adaptive immune responses. Kupffer cells and infiltrating tissue monocytes amplify inflammation by releasing cytokines and chemokines, while natural killer cells and T lymphocytes promote hepatocyte apoptosis and fibrosis. Obesity-related adipose tissue inflammation further contributes pro-inflammatory adipokines that affect the liver. The spleen-liver axis enhances this immunometabolic signaling: maladaptive changes in splenic immune reservoirs mobilize pro-inflammatory monocytes and activated T cells to the liver, while splenic macrophages secrete cytokines that worsen hepatic inflammation and fibrosis. Clinically, splenomegaly and splenic immune hyperactivity often coincide with disease progression [27], highlighting the role of the spleen-liver axis in MASLD development.

Intestinal dysbiosis and barrier dysfunction lead to lipopolysaccharides and microbial products crossing into the portal circulation, which then stimulate hepatic toll-like receptor pathways and cause inflammation. In addition to changes in microbiota composition, microbial metabolites also exert direct pathogenic effects: short-chain fatty acids influence insulin sensitivity and lipogenesis; trimethylamine-N-oxide (TMAO) impacts atherosclerosis and metabolic signaling; and endogenous ethanol production promotes oxidative stress and hepatocyte cell death. Abnormal bile acid metabolism and faulty choline use further worsen steatosis and inflammation [28].

Dietary patterns high in saturated fats, simple sugars, and ultra-processed foods promote lipogenesis and oxidative stress, while physical inactivity worsens insulin resistance [25]. Conversely, regular physical activity enhances insulin sensitivity, increases fatty acid oxidation, and decreases hepatic steatosis, serving as a protective behavioral factor. Alcohol consumption, even at modest levels, synergizes with metabolic stress to accelerate disease progression [29]. Environmental pollutants and endocrine-disrupting chemicals, including phthalates and bisphenols [30,31], accumulate in fatty liver tissue and contribute to metabolic dysregulation. Social determinants of health, such as urbanization, socioeconomic status, circadian rhythm disruption, and limited access to healthy foods, further influence the global burden and outcomes of MASLD [32,33].

Taken together, these interconnected areas form a multidimensional framework for understanding MASLD. This expanded model highlights not only the biological factors behind steatosis and fibrogenesis but also the social and environmental factors that critically influence disease presentation and clinical course. The pathogenesis of MASLD/MASH and its sequelae is briefly summarized in Figure 1.

## 3. Role of Diet and Lifestyle in MASLD

The development of MASLD results from the interaction between genetic and epigenetic factors with various acquired insults, including disordered liver lipid metabolism, systemic insulin resistance, hormonal and cardiovascular-related conditions, immune-inflammatory responses, changes in the gut–liver connection, and environmental and lifestyle factors. Among these, diet and lifestyle are the main environmental influences. Weight gain typically occurs when there is an energy imbalance caused by increased caloric intake and decreased caloric expenditure. The latter is heavily affected by sedentary behaviors and insufficient physical activity. Eating a diet high in calories, especially one rich in saturated fats, simple sugars, and fructose, further promotes fat buildup in the liver and metabolic damage.

## 4. Role of Weight Loss

Lifestyle interventions and weight loss are currently the primary treatments for MASLD [34,35,36], as only two pharmacological agents have received approval from regulatory agencies [19,20]. Sustained weight reduction directly enhances the histopathological features of MASH. Observational studies have consistently shown the positive effects of weight loss, as indicated by imaging biomarkers, although relatively few randomized clinical trials have reported histological endpoints.

Weight loss improves liver biochemical tests, histology, serum insulin levels, and quality of life in patients with MASLD [21,37,38,39,40,41]. A meta-analysis revealed that a minimum of 5% weight loss significantly improved disease severity, while a 7% reduction in body weight was associated with an improved NAFLD activity score (NAS) [37]. A prospective trial demonstrated greater reductions in NAS, MASH resolution, and fibrosis regression with more than 10% weight loss [21]. These findings support a dose–response relationship, where greater weight loss is associated with more significant improvements in liver inflammation, ballooning, and histological resolution of MASLD or MASH [42].

Caloric restriction (CR), independent of physical activity, consistently improves liver enzymes, hepatic inflammation, and fibrosis [21]. Even moderate weight loss through CR is beneficial, with a significant reduction of approximately 4.5% that improves steatosis, waist circumference, and serum ALT, AST, and lipid profiles [43]. A systematic review and meta-analysis of patients with Class III obesity (BMI ≥ 40 kg/m^2^) showed that energy-restricted diets resulted in a weight loss of at least 10% when observed for six weeks or longer [44]. When CR is combined with behavioral programs, very-low-energy diets achieve greater weight loss than behavioral interventions alone [45]. Interestingly, sex-specific outcomes have been observed, with women experiencing greater reductions in fat-free mass, hip circumference, and LDL cholesterol compared to men [46].

Guidelines recommend structured physical activity and dietary caloric restriction (Table 1). Both aerobic and resistance training, including intensive lifestyle intervention (ILI), are effective [35].



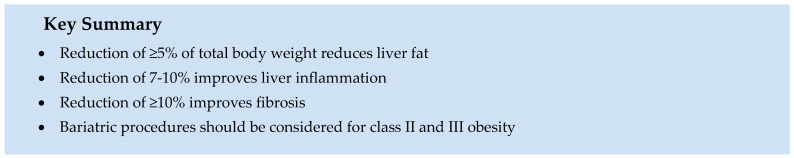



## 5. Role of Dietary Modifications

Beyond total caloric restriction, the quality and distribution of dietary intake are important factors affecting MASLD risk and progression. Excessive calorie intake, especially within a Western-style diet, promotes gut dysbiosis, inflammation, and obesity [52]. Gaining as little as 3–5 kg can increase the likelihood of developing MASLD [53]. Additionally, eating habits are significant: frequent snacking between meals has been independently associated with higher levels of hepatic steatosis [54].

The Western diet, characterized by low consumption of fruits, vegetables, whole grains, and fish, leads to deficiencies in fiber, vitamins, minerals, and antioxidants [55,56]. In contrast, healthy dietary patterns that include fruits, vegetables, nuts, olive oil, and fish exhibit protective associations [56,57]. A retrospective study revealed that individuals with high adherence to a Western pattern were twice as likely to have MASLD compared to those with low adherence, regardless of age, gender, BMI, physical activity, and energy intake [56]. Similarly, high adherence to a Western diet has been associated with an increased risk of fibrosis [58]. Conversely, participants with the highest adherence to healthy dietary patterns were 41% less likely to develop MASLD [56]. Experimental evidence supports this further, as a Western-style, low-choline, high-sugar, high-fat diet has been shown to cause steatosis, inflammation, and fibrosis in mice [59].

Relying heavily on processed foods also leads to high salt intake [60]. This has been linked to chronic metabolic inflammation [61,62], which promotes the progression of MASLD. Although the role of salt in preventing or treating MASLD remains unclear, its well-known connection to hypertension and cardiovascular disease suggests an increased risk of morbidity and mortality [63,64].

Taken together, caloric restriction remains the most well-supported approach for improving MASLD. However, enhancing diet by cutting saturated fats, simple sugars, and excess fructose, while following a Mediterranean-style eating pattern, further boosts metabolic and liver health.

### 5.1. Role of Macronutrients

The components of macronutrients in a diet are linked to the development of MASLD, regardless of energy intake [65]. Unhealthy eating habits can directly promote MASLD/MASH by affecting triglyceride (TG) buildup in the liver, altering insulin sensitivity, and changing postprandial TG metabolism. Patients with MASH tend to consume more saturated fats and fewer polyunsaturated fatty acids (PUFAs), fiber, and vitamins C and E [66,67]. An extensive population-based study conducted by Rotterdam found that eating a diet high in animal protein was closely associated with MASLD in individuals with abnormal fat accumulation [68]. Regardless of weight loss and calorie intake, maintaining a balanced diet with the correct macronutrient composition can help reduce liver fat buildup. Therefore, understanding and analyzing specific macro- and micronutrients, as well as their effects on the liver, is crucial [69].

#### 5.1.1. Fats

Lipids are essential for membrane integrity, hormone production, and energy metabolism, but their impact on MASLD varies depending on fatty acid composition [70,71].

##### Saturated Fats

Saturated fatty acids (SFAs) are present in animal products (e.g., dairy products, red meat, butter, whole milk), plant-based products (e.g., coconut oil, palm oil), and processed foods (e.g., desserts and sausages). SFAs, especially palmitic and lauric acids, raise intrahepatic triglyceride content (IHTG) and plasma ceramides more than PUFAs and free sugars by promoting adipose tissue lipolysis [72]. SFAs also reduce insulin sensitivity by inducing endotoxemia, contributing to lipotoxicity signaling, and speeding up inflammation and fibrogenesis [73,74,75,76,77,78].

##### Monounsaturated Fats

Monounsaturated fatty acids (MUFA) are mainly found in olive oil, avocados, and nuts [79]. MUFAs are less toxic to cells than saturated fatty acids (SFAs) and can even reduce SFA-related cellular damage [80]. Consuming MUFA is linked to a healthier lipid profile, characterized by lower LDL cholesterol, triglycerides, a decreased total cholesterol-to-high-density lipoprotein (HDL) ratio, as well as better insulin sensitivity and reduced hepatic steatosis [79,80,81,82,83,84,85].

##### Polyunsaturated Fats

There are two main types of essential PUFAs: omega-3 and omega-6. Omega-3 PUFAs are commonly found in cold-water and marine fish (e.g., salmon, mackerel, tuna, herring, sardines), flaxseed, chia seeds, and walnuts [86]. Omega-6 PUFAs are usually present in various vegetable oils (e.g., safflower, canola, sunflower, corn, soybean), sunflower and pumpkin seeds, corn, Brazil nuts, and walnuts [86]. The primary dietary omega-6 PUFA is linoleic acid. Omega-3 PUFAs include alpha-linolenic acid (ALA), eicosatetraenoic acid (EPA), and docosahexaenoic acid (DHA) [86,87].

Omega-3 PUFAs (EPA, DHA, ALA) consistently reduce hepatic steatosis, improve metabolic parameters, and exert anti-inflammatory and antifibrotic effects, although fibrosis regression remains inconsistent across RCTs [86,87,88,89,90,91,92,93,94,95]. Notably, protective benefits may be most relevant in high-risk groups, including women [89,90,91,95]. A balanced intake with omega-6 fatty acids appears essential for optimal metabolic outcomes [86,89].

##### Trans Fats

Trans fats are found in partially hydrogenated vegetable oils, desserts, cream, or solid fats and are a recognized risk factor for developing MASLD [68]. Industrial trans fats have strong pro-oxidative and pro-inflammatory effects, contributing to obesity, insulin resistance, and MASLD progression in experimental models [68,96,97,98,99,100]. Although clinical evidence is limited, it supports restricting their intake.

#### 5.1.2. Carbohydrates

Carbohydrate quality, rather than total intake, is key to MASLD development. Excess refined carbs and added sugars, especially fructose, trigger hepatic de novo lipogenesis (DNL), increase visceral fat, worsen insulin sensitivity, and activate inflammatory and fibrogenic pathways [101,102,103,104,105,106,107,108,109,110,111,112]. High intake of sugar-sweetened drinks is strongly linked to MASLD occurrence and progression, with a dose–response effect seen in long-term studies [113,114,115,116,117,118]. Conversely, diets focused on whole grains, fiber-rich foods, fruits, and vegetables enhance insulin sensitivity, lower intrahepatic triglycerides, and reduce metabolic risk [105,106,107].

##### Fructose

Fructose strongly stimulates de novo lipogenesis [108], leading to SFAs associated with MASH [77]. It can induce lipogenesis by upregulating lipogenic gene expression, such as SREBP-1c and ChREBP, which increase the FFA pool in the liver by being metabolized through energy-mediated processes into triglycerides, promoting DNL, and causing depletion of adenosine triphosphate (ATP). It is also linked to bacterial overgrowth in the small bowel and elevated endotoxin levels in the portal vein, which can promote inflammation and progression from steatosis to MASH [108,109,110,111,112,113,114,115,116,117,118].

#### 5.1.3. Dietary Fiber

A lack of fiber in the diet has been linked to MASLD regardless of race or ethnic group [119,120]. The mechanism behind this association remains unclear [121,122]; however, a recent study combining dietary fiber and probiotic therapy found that this combined approach modulates fatty acid oxidation by activating the Acly/Nrf2/NF-κB signaling pathway. It also reduces inflammation and lipid synthesis [123]. Among fibers, prebiotic fibers are non-digestible carbohydrates (e.g., garlic, asparagus, and onions) that primarily function by regulating the gut microbiota [121]. 

The gut microbiota primarily helps with nutrient absorption in the liver. It also influences hepatic inflammation by providing toll-like receptor ligands (TLRs), which cause hepatocytes to release pro-inflammatory cytokines. Disruption of these functions leads to impaired lipid and glucose balance, resulting in steatogenesis, inflammation, and fibrosis, indicating the development of MASH [122]. Altering the gut microbiota with prebiotic fibers may offer a therapeutic approach to treating MASH.

Fructooligosaccharides promote their prebiotic effects by encouraging the growth of Bifidobacteria in the large intestine, which helps prevent the proliferation of harmful bacteria [124]. Viscous dietary fiber intake decreases insulin resistance (IR), adiposity, and hepatic steatosis regardless of fermentability [125]. A randomized, placebo-controlled clinical trial showed that supplementing with oligofructose (a dietary fiber found in vegetables) increased Bifidobacterium growth and notably improved hepatic steatosis and NAS compared to placebo-induced weight loss [126]. A recent meta-analysis revealed improvements in BMI, ALT, AST, and insulin resistance with fiber supplementation in MASLD [127].

#### 5.1.4. Proteins

Proteins are essential macronutrients for enzymatic activity, cell repair, and energy regulation. Besides quantity, the kind and source of protein significantly affect MASLD risk and progression.

High intake of animal protein, especially red and processed meats, has been linked to MASLD, type 2 diabetes, and metabolic syndrome, while plant-based proteins appear protective. Replacing animal protein with plant protein significantly reduces MASLD risk [105,128]. Meta-analyses show a positive link between red meat consumption and MASLD development [129], with population studies indicating increased mortality risk from both processed and unprocessed red meat, which is decreased when replaced with white meat [130]. Cooking red meat at high temperatures also contributes to insulin resistance through heterocyclic amine formation [131].

In contrast, fish, eggs, white meat, and low-fat dairy offer high-quality protein with a lower metabolic risk. Fish and eggs also provide omega-3 fatty acids and choline, nutrients essential for hepatic lipid export and anti-inflammatory pathways. Choline deficiency hampers very-low-density lipoprotein (VLDL) secretion, leading to intrahepatic fat buildup and faster fibrosis, highlighting the importance of choline-rich foods in MASLD prevention.



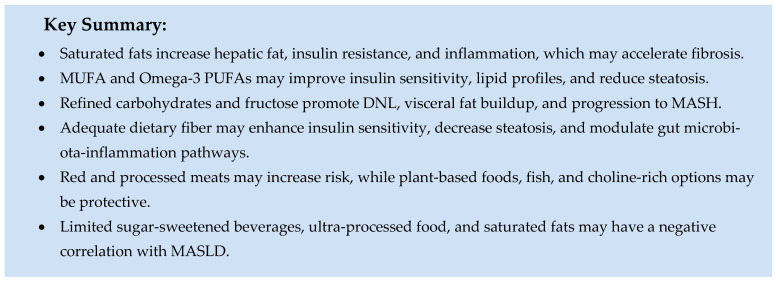



### 5.2. Role of Micronutrients

Micronutrients are the vitamins and minerals necessary for most body functions, disease prevention, and overall health. The different vitamins and minerals, along with their effects on MASLD, are discussed below. 

#### 5.2.1. Vitamins

Vitamins with antioxidants support health by reducing oxidative stress [132]. This property may help reverse hepatic fibrosis in patients with MASLD, slow the progression of liver injury, and prevent MASH from advancing to a more severe stage [133].

##### Vitamin E

Vitamin E is a fat-soluble vitamin with antioxidant, anti-inflammatory, and anti-apoptotic properties [134,135], with the most evidence supporting its therapeutic benefits in MASLD. It enhances its anti-inflammatory effect by increasing adiponectin levels and reducing the release of various inflammatory cytokines such as TNF-α, IL-1, IL-2, IL-4, and IL-8 [136]. It neutralizes hydroxyl, peroxyl, and superoxide radicals and protects against plasma lipid and LDL peroxidation. Vitamin E can scavenge reactive oxygen species and also eliminate reactive nitrogen species [137]. Additionally, this vitamin acts as an anti-apoptotic agent by increasing the expression of the anti-apoptotic protein B-cell lymphoma-2 (BCL-2) and decreasing the levels of pro-apoptotic proteins BAX and p53 [135].

According to the PIVENS (Pioglitazone versus Vitamin E versus Placebo for Treatment of NASH) trial, daily supplementation of vitamin E (800 IU) was considered more effective than a placebo for treating MASH in non-diabetic adults [138]. Data from this study showed improvements in MASH histological parameters. There was no difference between pioglitazone and placebo. However, treatment with vitamin E also resulted in a decline in MASH, with improved ALT levels that reduced hepatic steatosis, lobular inflammation, and hepatocellular ballooning. No significant reduction in fibrosis was observed [138]. This study, however, only included patients with MASH who did not have diabetes. Since insulin resistance and progression to T2DM and metabolic syndrome are key aspects of MASH development, the results of the PIVENS trial cannot necessarily be generalized to this patient group. The effects of vitamin E were further studied in the diabetic population. A cohort of T2DM patients participated in a randomized, double-blind, placebo-controlled trial where they received either vitamin E 400 IU twice daily, vitamin E 400 IU twice daily plus pioglitazone 45 mg/day, or a placebo. The data showed that combination therapy (vitamin E 400 IU twice daily plus pioglitazone 45 mg/day) was more effective than placebo in improving liver histology in MASH and T2DM patients. However, vitamin E alone was ineffective at altering the primary histological abnormalities. As with the PIVENS trial, fibrosis was not improved [139].

Daily supplementation with vitamin E (mainly α-tocopherol, at least 200 IU daily) may lower hepatic biomarkers of lipid peroxidation and boost the activity of natural antioxidants in the liver [140]. A systematic review and meta-analysis of 12 randomized clinical trials showed improvements in ALT and AST levels in the vitamin E group compared to placebo [141]. Another meta-analysis of high-quality RCTs assessed how vitamin E treatment affected patients with MASH compared to metformin and thiazolidinediones. This analysis found that vitamin E significantly enhanced histological scores for steatosis, lobular inflammation, and ballooning [142]. A prospective study comparing vitamin E (400 IU twice daily) and pentoxifylline (400 mg three times daily) with vitamin E alone demonstrated greater fibrosis regression and better insulin resistance improvements with combination therapy [143]. Another recent meta-analysis concluded that vitamin E might improve the biochemical, metabolic, and histological features of MASLD [144]. Vitamin E appears to be a promising treatment option for MASLD; however, larger prospective trials are necessary to confirm this relationship. According to the AASLD and EASL-EASD-EASO, vitamin E (800 IU/day) may be used in nondiabetic adults with biopsy-confirmed MASH.

##### Vitamin C

Vitamin C has antioxidant properties and can neutralize free radicals [145], similar to vitamin E. It may lower hepatic oxidative stress and inflammatory markers [146,147]. Regulation of adiponectin by vitamin C reduces inflammation, hepatic lipid buildup, and systemic IR through hepatic lipid balance [148]. In rats with dexamethasone-induced glucose intolerance, vitamin C supplementation was associated with improved insulin resistance [149]. A cross-sectional study found that low vitamin C intake in adults could increase the risk of developing MASLD [150]. Conversely, similar vitamin C levels were observed in both MASLD and non-MASLD (healthy controls) subjects [151]. Combined dietary intake of vitamin C and E has been inversely related to MASLD severity [152]. In a prospective, double-blind, randomized, placebo-controlled trial, a group of patients with MASH received either 1000 IU and 1000 mg of vitamin C and E or a placebo daily for six months. Combination therapy with vitamins C and E showed a significant improvement in MASLD fibrosis scores [153]. However, this may not reflect the efficacy or safety of using vitamin C alone. Another study with lower doses of vitamins C and E demonstrated improvements in ALT levels, necroinflammatory activity, and fibrosis [154]. More recently, a double-blind randomized clinical trial using vitamin C alone reported improvements in the treatment group’s AST, ALT, fasting insulin, and fasting glucose [155].

##### Vitamin D

Vitamin D is an essential nutrient that plays a key role in calcium balance. Emerging evidence suggests that vitamin D deficiency is linked to MASLD [156,157,158]. Although another study found no connection between vitamin D deficiency and MASLD, among those with MASLD, having sufficient vitamin D levels was associated with a lower risk of liver fibrosis in a dose-dependent way [159]. When taken as a supplement, vitamin D may protect against fibrosis by inhibiting HSC proliferation [156] or by improving inflammatory markers [160]. 

Vitamin D influences insulin resistance by binding directly to vitamin D receptors (VDR) on pancreatic beta cells to enhance insulin secretion. A deficiency in vitamin D may lead to insulin resistance (IR) and raise the risk of MASLD [161]. An RCT with 162 patients showed improved IR within the vitamin D treatment group compared to the control [162]. Another double-blind RCT found that calcitriol was 1.8 times more effective than cholecalciferol in reducing insulin resistance in MASLD patients [163]. Vitamin D also impacts the immune system by regulating genes involved in innate immunity. Toll-like receptors 2 and 4 on macrophages, polymorphonuclear cells, monocytes, and epithelial cells become activated when vitamin D levels are low, playing a key role in MASLD development [164]. Vitamin D promotes hepatic autophagy, which helps prevent hepatic steatosis [165]. Currently, several clinical trials are exploring vitamin D’s effects on liver abnormalities and its potential benefits for MASLD patients. A recent meta-analysis found no significant impact of vitamin D on MASLD [166]. However, other studies report reductions in liver steatosis and fibrosis after one year of daily supplementation with 1000 IU of vitamin D [167], as well as improvements in lipid profiles and liver enzyme levels after six weeks of taking 2000 IU daily [168].

##### Vitamin A

Vitamin A, also called retinol or retinoic acid, regulates cellular processes such as cell growth and immune function. An inadequate intake or vitamin A deficiency promotes the progression of MASLD [169], where quiescent HSCs store dietary retinol and retinyl-palmitate esters (RE) inside lipid droplets. When activated, HSCs rapidly lose their vitamin A content in response to liver injury. This process has been linked to the onset of vitamin A deficiency, which worsens chronic liver disease [170]. Retinol released from HSCs is partly converted into retinoic acid, which can modulate the immune response by decreasing liver injury severity and promoting liver regeneration. If hepatic damage continues, activated HSCs gradually transform into myofibroblast-like cells that produce extracellular matrix, leading to liver fibrosis. Retinoic acid has been shown to reduce the proliferation and spread of hepatocellular carcinoma [170]. Multiple studies suggest that preventing MASLD depends on proper activation of retinoic acid signaling. Serum retinol levels are low in patients with the MASLD risk variant of PNPLA3. Despite this evidence and numerous studies highlighting vitamin A’s benefits on liver lipid metabolism in obesity-related MASLD animal models [171], no clinical trials are currently exploring its therapeutic potential in humans. A study linking total dietary vitamin A intake to MASLD risk found an inverse relationship, particularly among adults under 45 and women [172].

##### Vitamin B3

Vitamin B3, also known as niacin or nicotinic acid, is water-soluble and easily obtained from dietary sources. It acts as a precursor to the coenzymes nicotinamide adenine dinucleotide (NAD) and nicotinamide adenine dinucleotide phosphate (NADP), which are essential in lipid metabolism [173]. Vitamin B3 may reduce intrahepatic triglycerides (IHTG) associated with MASLD by inhibiting diacylglycerol acyltransferase 2, an enzyme that catalyzes the final step in triglyceride synthesis [174]. Nicotinamide (NAM), a precursor to methyl nicotinamide and a form of niacin, has been shown in studies to improve insulin resistance caused by a high-fat, high-fructose diet in rats when delivered in chitosan nanoparticles [175]. NAM also helps prevent liver steatosis and fibrosis by regulating redox potential through glucose-6-phosphate dehydrogenase- and malic enzyme-dependent mechanisms [176]. Some research suggests that higher dietary intake of niacin [177] or niacin supplementation [178] may be associated with lower liver fat levels. However, an RCT found that while niacin treatment improved serum triglycerides, VLDL, and insulin sensitivity, it did not reduce hepatic fat accumulation [179]. Furthermore, some studies indicate that niacin might induce insulin resistance [180].

##### Vitamin B6

Vitamin B6, also known as pyridoxine, is water-soluble, found in various foods, and essential for normal brain development. In a cross-sectional study, vitamin B6 is positively associated with the development of hepatic steatosis [181]. However, a recent open-label, single-arm, single-center study examined the therapeutic effects of vitamin B6 supplementation (90 mg daily) in patients with MASLD. Vitamin B6 supplementation has been shown to significantly reduce hepatic fat accumulation [182]. Pyridoxamine (vitamin B6) influences oxidative stress, AGE products, and TNF-α levels [183]. 

##### Vitamin B9

The liver is a vital organ that stores and metabolizes vitamin B9, also known as folate or folic acid, found in various foods [184]. Folic acid supplementation inhibits hepatic lipogenesis, helping to reduce hepatic steatosis in high-fructose-fed rats by activating liver kinase B1, AMP-activated protein kinase, and acetyl-CoA carboxylase in the liver [185]. It also significantly decreases the pro-inflammatory NF-κB pathway and cytokine expression in mice [186]. Another study examined folate receptor beta (FR-β) protein expression in human MASLD and rodent models of MASH, finding increased FR-β expression in both, making it a potential future therapeutic target. Folate supplementation was associated with reduced levels of pro-inflammatory cytokines TNF-α and CXCL8 (chemokine (C-X-C motif) ligand 8), decreased LC3B (light chain 3 B) expression, and increased IL-22 levels in a dose-dependent manner [187]. Low vitamin B9 levels have been linked to increased severity of MASH [188]. A retrospective study in a Chinese population identified low serum folic acid as an independent risk factor for MASLD [189]. A recent randomized controlled trial assessing 1 mg daily oral folic acid supplementation found no significant changes in liver enzymes, lipid profiles, insulin resistance, or hepatic steatosis grade [190].

##### Vitamin B12

Vitamin B12, or cobalamin, exists in two metabolically active forms in humans: methylcobalamin and 5′-deoxyadenosylcobalamin. The other two forms, cyanocobalamin and hydroxocobalamin, become biologically active once converted into these forms. They serve as cofactors for the mitochondrial enzyme methylmalonyl-CoA mutase, which is involved in lipid metabolic pathways [191]. Vitamin B12 is stored in the liver. A systematic review and meta-analysis found that vitamin B12 levels are not associated with MASLD. However, a significant difference in homocysteine levels was observed in MASLD patients, suggesting that homocysteine could be a potential marker of liver damage [192]. Low vitamin B12 levels have been linked to increased severity of MASH [188]. An RCT assessing 1 mg of daily vitamin B12 supplementation versus a placebo in patients with MASLD reported a significant reduction in homocysteine levels in the treatment group compared to the placebo. The study also noted a significant within-group decrease in liver steatosis and fasting blood glucose levels in the treatment group; however, these differences were not seen when comparing between groups [193].



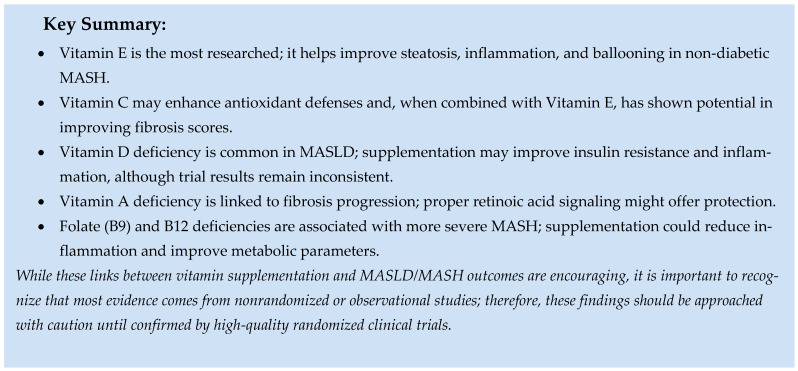



### 5.3. Role of Minerals

#### 5.3.1. Calcium and Phosphorus

Disruption of calcium and phosphorus balance is increasingly recognized as a key factor in MASLD. Altered calcium signaling impairs lipid management, increases oxidative stress, and promotes inflammation and fibrosis, while balanced calcium flux supports lipid breakdown, autophagy, and hepatocyte regeneration [194]. Similarly, phosphorus influences liver lipid metabolism by stimulating autophagy, enhancing fatty acid oxidation, and reducing triglyceride accumulation, with optimal levels providing protective effects. Both calcium and phosphorus play a dual role: maintaining homeostatic levels helps sustain metabolic stability and prevents fat accumulation, whereas deficiency or excess can lead to steatosis, inflammation, and progression to advanced liver disease [195]. Therefore, maintaining balanced mineral levels is essential for preventing and influencing the progression of MASLD.

#### 5.3.2. Zinc and Magnesium

Zinc and magnesium both play crucial roles in the development of MASLD. Zinc is essential for lipid metabolism, antioxidant defense, and insulin signaling. Deficiency impairs lipophagy, increases oxidative stress, disrupts the unfolded protein response, and can lead to liver cancer and fibrosis, while sufficient levels protect against steatosis, inflammation, and insulin resistance. However, paradoxically, elevated circulating zinc at the onset may indicate dysfunctional liver utilization or increased intake.

Magnesium, on the other hand, influences insulin sensitivity, mitochondrial functions, and inflammatory signaling. Deficiency impairs tyrosine kinase activity and glucose uptake, increases oxidative stress, and leads to hepatocyte ballooning and steatohepatitis. Conversely, higher intake improves insulin resistance, reduces inflammation, and lowers the long-term risk of MASLD. However, excessive dietary calcium, by altering absorption dynamics, may diminish the effects of magnesium.

Together, zinc and magnesium affect MASLD progression through common mechanisms involving oxidative stress, insulin resistance, and fibrogenesis. The levels of both elements are usually harmful when deficient, and supplementation provides protective effects in both experimental and human studies [196].

#### 5.3.3. Iron and Selenium

Iron and selenium homeostasis are better understood as modulators of MASLD pathogenesis; their impact, whether protective or adverse, depends on circulating levels of these elements. Increased iron levels are consistently associated with a higher risk of MASLD through mechanisms involving disrupted iron transport, oxidative stress, and resulting insulin resistance and subclinical inflammation. Elevated liver iron promotes hepatocellular damage by producing reactive oxygen species, activating stellate cells, and increasing fibrosis. It also contributes to systemic disease via β-cell dysfunction and cardiovascular injury. In contrast, moderate iron restriction reduces liver damage, emphasizing the importance of maintaining an optimal iron balance.

Selenium also shows dose dependence related to MASLD. While at normal selenium and selenoprotein levels, it provides antioxidant and immunomodulatory benefits. However, excessive selenium intake increases production of reactive oxygen species via selenomethionine metabolites, leading to oxidative stress, inflammation, and lipid buildup in the liver. Studies in mice and humans confirm that high selenium exposure accelerates liver injury and metabolic issues, and correcting selenium excess may help in clinical settings.

Together, the studies demonstrate that both iron and selenium serve as double-edged regulators in MASLD, where homeostatic balance provides protection, whereas excess promotes disease progression [197].



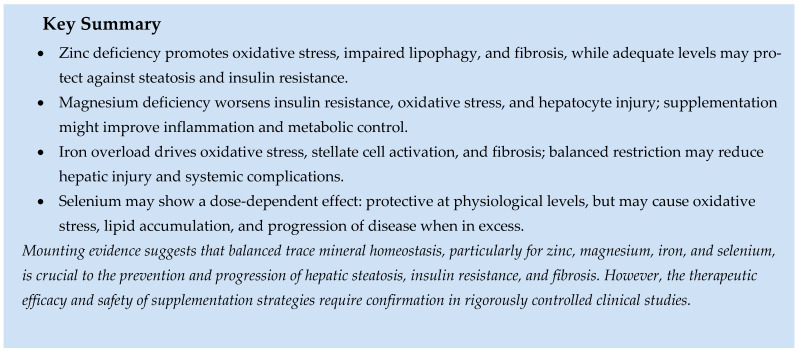



### 5.4. Role of Herbal Supplements

#### 5.4.1. Milk Thistle

Silymarin is a plant seed extract with antioxidant, anti-inflammatory, and antifibrotic properties. This extract is a complex mixture of six significant flavonolignans and other minor polyphenolic compounds derived from the plant Silybum marianum [198]. The therapeutic effects of silymarin on MASLD have been extensively studied. Silymarin’s hepatoprotective effects are attributed to its antioxidant activity, modulation of inflammatory pathways, antifibrotic properties, and regulation of lipid and glucose metabolism [199]. Its primary active component, silybin, interacts with nuclear receptors such as the farnesoid X receptor (FXR), influencing bile acid metabolism and insulin sensitivity [200]. Multiple studies have shown improvements in AST and ALT levels [201,202], and some enhancement of liver fibrosis with silymarin supplementation [203]. However, in a randomized trial of 99 patients, silymarin supplementation (700 mg, given three times daily for 48 weeks) did not reduce NAS by 30% or more in a significantly higher proportion of patients with MASH compared to placebo [198]. A randomized, double-blind, placebo-controlled trial involving 78 patients also found no significant reduction in NAS with silymarin therapy compared to placebo [204]. A recent systematic review and meta-analysis of 26 RCTs reported significantly decreased levels of total cholesterol, triglycerides, LDL, and HOMA-IR, along with increased HDL levels with silymarin administration. Additionally, they observed reductions in AST, ALT, fatty liver score, and hepatic steatosis with silymarin supplementation [205].

#### 5.4.2. Turmeric

Turmeric belongs to the ginger family. Its primary biologically active component is curcumin. It is thought to offer medicinal benefits due to its strong antioxidant, anti-inflammatory, and metabolic regulatory properties [206,207]. Curcumin can help reduce inflammation by modulating nuclear factor kappa B (NF-κB), a group of transcription factors that control genes involved in various inflammatory responses. This is significant because inflammation plays a key role in the development of steatohepatitis [208]. Furthermore, curcumin improves gut health by encouraging the growth of beneficial bacteria and releasing antioxidant, anti-inflammatory, and anti-tumor metabolites during metabolism [209]. Research indicates that curcumin decreases insulin resistance (IR) in mice fed a high-fat diet [210]. One study reported notable improvements in glucose disposal in the liver and fat tissue with curcumin supplementation [211]. Another found that oral curcumin inhibits fat tissue lipolysis, which reduces free fatty acid flow toward the liver and thus reduces hepatic IR [210]. A recent meta-analysis revealed significant improvements in fasting blood glucose, insulin resistance, triglycerides (TG), total cholesterol, LDL cholesterol, weight, and BMI [212]. A different meta-analysis showed similar results, including better fasting blood glucose levels, reduced insulin resistance, decreased waist circumference, and lower serum levels of ALT, AST, total cholesterol, and LDL. However, serum levels of TG, LDL, HbA1c, body weight, and BMI remained unchanged with curcumin supplementation [213].

#### 5.4.3. Garlic

Garlic is a commonly used herb in cooking. The main active compound is S-allymercaptocysteine (SAMC), an antioxidant that reduces inflammation [214,215], as shown in animal studies. A meta-analysis of four studies found that garlic supplementation was associated with improvements in ALT, AST, total cholesterol, LDL cholesterol, triglycerides, and fasting blood sugar levels compared to a placebo [216]. Another meta-analysis reported similar results, with significantly lower levels of ALT, AST, LDL cholesterol, and total cholesterol after garlic supplementation [217].

#### 5.4.4. Basil, Lavender, Peppermint, Oregano, and Rosemary

Most aromatic herbs, such as rosemary, peppermint, basil, lavender, and oregano, contain ursolic acid [218]. Carnosic acid is found explicitly in rosemary [219]. In animal studies, both of these compounds demonstrate anti-inflammatory and antioxidant properties [218,220]. However, there is a lack of definitive human studies confirming similar clinical benefits. 

#### 5.4.5. Ginger

Ginger has anti-lipogenic, anti-inflammatory, and antioxidant properties, as observed in animal studies [221]. An RCT involving 44 patients who received either two grams of ginger supplement daily or a placebo found that the ginger group experienced significant improvements in ALT, inflammatory cytokines, insulin resistance, and hepatic steatosis grade compared to the placebo group [222]. A meta-analysis of 17 in vivo experiments and three clinical trials concluded that ginger supplementation was linked to improvements in total cholesterol, LDL cholesterol, HDL cholesterol, TG, ALT, and AST levels [223]. However, there is not enough data to suggest any significant clinical benefits of MASLD.

#### 5.4.6. Gingko Biloba

Ginkgo Biloba reduces oxidative stress, improves liver enzyme levels, and decreases hepatic steatosis and inflammation, as shown in animal studies [224,225]. Ginkgo biloba extract was associated with better insulin resistance (IR), glucose intolerance, lipid accumulation, and hepatic steatosis in the high-fat diet mouse model [226].

#### 5.4.7. Ginseng

Ginseng is a herb with antioxidant and anti-inflammatory properties that protect against steatosis, hepatic inflammation, and fibrosis by modulating liver enzyme levels in MASLD, as seen in animal studies [227,228,229]. Other traditional Chinese herbs like Goji Berry, Lotus, Astragalus, and Ciruwujia are being studied in animal models, but clinical studies have not been conducted with these herbs.

#### 5.4.8. Licorice

Licorice has been linked to improved levels of AST and ALT and is suggested to support lipid balance [230]. In an RCT, women who took 1 g of licorice root powder daily for 12 weeks showed significant improvements in ALT, insulin, insulin resistance, and ultrasonographic liver steatosis compared to the placebo group [231].

#### 5.4.9. Rosa Damascena, Plantago Major

Rosa damascena is a plant cultivated in Iran and traditionally used in Persian herbal medicine. This herbal remedy is thought to have hepatoprotective effects due to its bioactive compounds, including phenolic acids and flavonoids. In a 12-week randomized controlled trial (RCT), where participants took 3 g of Rosa damascena capsules, there was a significant reduction in ALT, lipid profile (except HDL), weight, BMI, waist circumference, diastolic blood pressure, and MASLD grade on ultrasound [232]. Plantago major is another herb with potential liver health benefits. In a randomized, double-blind, placebo-controlled clinical trial, patients received 2 g of Plantago major seeds compared to a placebo and were monitored for 12 weeks. Results showed a significant decrease in serum levels of ALT, AST, TG, and waist circumference in the treatment group compared to the control group [233].

#### 5.4.10. Berberine

Berberine is an isoquinoline alkaloid derived from medicinal plants like Berberis. Berberine has been suggested to help prevent gut microbiota-derived lipopolysaccharide (LPS)-induced intestinal barrier dysfunction and reduce inflammation in metabolic diseases [234,235]. This may improve glucose and lipid metabolism. In a meta-analysis, Berberine showed improvements in body weight, HOMA-IR, AST, ALT, GGT, total cholesterol, and LDL-C [236]. Additionally, Berberine has been shown to have anti-HCC effects by regulating the cell cycle, promoting autophagy, and inducing cell death [237].



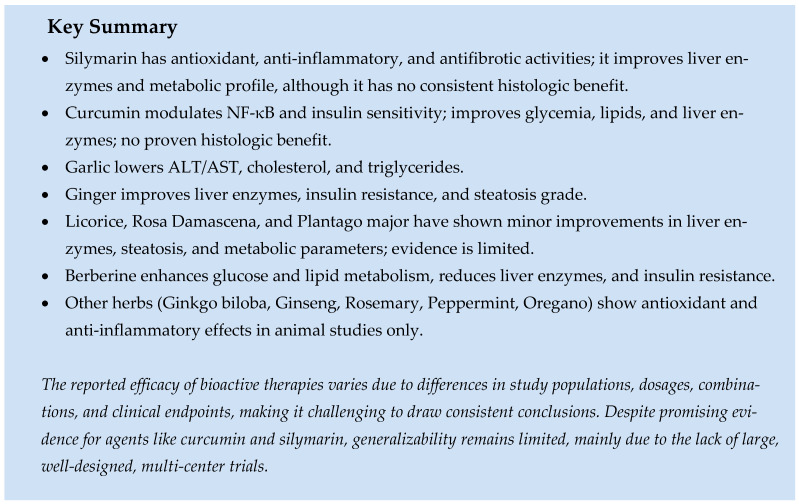



### 5.5. Role of Other Supplements

#### 5.5.1. Probiotics

Imbalance in the gut microbiome is associated with the release of bacterial endotoxins and the production of metabolites, which can contribute to obesity and MASLD. Changes in the intestinal microbiota can affect the liver by transferring microbial products and absorbing metabolites [238,239]. This leads to the endogenous production of ethanol, activation of inflammatory cytokines through lipopolysaccharides, and changes in choline and bile acid metabolism [124].

Probiotics contain various beneficial bacteria (e.g., *Bifidobacterium* and *Lactobacillus* spp.) and fungi (e.g., *Saccharomyces* spp.). The bioactive components produced by these live microorganisms benefit human health [240]. According to a study by Mohamad et al., probiotic supplementation with *Lactobacillus* and *Bifidobacterium* species reduced intestinal permeability, thereby decreasing blood endotoxin levels and supporting MASLD treatment [241].

In a double-blind clinical RCT, consuming a tablet containing 500 million *Lactobacillus bulgaricus* and *Streptococcus thermophilus* bacteria improved serum AST and ALT levels in patients with MASLD [240]. The study found that probiotics lowered IHTG and serum AST levels but did not cause significant changes in anthropometric measures compared to standard care. Patients with MASLD who took *Bifidobacterium longum* with fructooligosaccharides alongside lifestyle modifications showed notable reductions in inflammatory markers, including TNF-α and CRP, as well as serum AST and endotoxins. These improvements also included reductions in insulin resistance, hepatic steatosis, and the MASH activity index [242]. A recent systematic review and meta-analysis of 41 RCTs by Rong et al. reported significant improvements in liver steatosis, AST, and ALT with probiotic treatment [243]. Additionally, an RCT demonstrated that probiotic yogurt intake in MASLD patients improved insulin resistance, ALT, and hepatic fat fraction [244].

Synbiotics, which combine prebiotics and probiotics, have also shown promise. In an RCT, consuming synbiotic yogurt improved liver enzyme levels and reduced hepatic steatosis in patients with MASLD [245]. However, current evidence does not confirm any benefits regarding liver fibrosis or slowing the progression of liver disease. 

#### 5.5.2. Caffeine

Several research studies examining the effects of coffee on patients with MASLD suggest that coffee may have a protective effect on MASLD. Coffee contains antioxidant and anti-inflammatory properties [246]. According to data from a meta-analysis on coffee and tea, it is suggested that coffee and tea consumption are less likely to be associated with the development of metabolic syndrome [247]. An improvement in levels of AST and ALT was reported in patients at risk for liver disease after coffee consumption and is inversely related to the progression of steatohepatitis in MASLD patients [248]. Drinking two regular cups of coffee daily has been linked to reductions in hepatic steatosis and fibrosis [246,249]. A study involving 1326 patients with MASLD and a median follow-up of 11.6 years found that those who consumed ≥2 and <3 cups of coffee per day had a significantly lower risk of developing advanced liver fibrosis [250]. Coffee consumption may also have chemoprotective effects. One study revealed that coffee was 44% more effective in reducing the risk of HCC in individuals who consumed more than three cups daily [251]. A meta-analysis of 13 studies on liver cancer risk concluded that there was a significant inverse correlation between coffee intake and the risk of liver cancer [252]. Furthermore, the benefits of coffee consumption could extend to gut microbiota, as it has been linked to an increase in *Bifidobacterium* spp., a known human probiotic [253,254]. While regular coffee intake is beneficial, recommendations should be personalized, considering other health conditions.

#### 5.5.3. Green Tea

Green tea has antioxidant and anti-inflammatory properties. The primary components responsible are epigallocatechin-3-gallate (EGCG) and polyphenols. Supplementing with green tea extract modulated liver enzymes in patients with MASLD [255]. It decreased hepatic steatosis and insulin resistance, as demonstrated in animal studies [256]. A meta-analysis found that green tea lowered levels of AST and ALT enzymes in patients with MASLD; however, in healthy individuals, a slight but significant increase in liver enzymes was noted [257]. An RCT investigating the effects of green tea extract on liver function and fat showed that supplementation reduced ALT levels and liver fat content. Nonetheless, it is important to mention that this RCT involved a small sample size of 17 individuals with MASLD [258].

#### 5.5.4. Low-Calorie Sweeteners

Low-calorie sweeteners (LCSs) are optional sugar substitutes approved by the FDA. Over the past twenty years, medical research has examined several safety concerns regarding their health risks, such as an increased risk of developing metabolic syndrome, type 2 diabetes, significant weight gain, cardiovascular disease, and disruption of the gut microbiome [259]. The American Heart Association and the American Diabetes Association recommend limiting the use of LCSs because there is no solid evidence about their effects on body weight and cardio-metabolic risk factors [260]. 

#### 5.5.5. Resveratrol

Resveratrol, a phytoalexin polyphenol chemically known as trans-3,5,4′-trihydroxystilbene, is found in red grapes and red wine, showing strong antioxidant and anti-inflammatory effects. Studies have demonstrated that resveratrol lowers oxidative stress, hepatic steatosis, and inflammation by activating the AMPK-SIRT1 pathways [261,262]. Some RCTs investigating the effect of daily resveratrol supplements reported improvements in AST, ALT, and insulin resistance [263,264]. However, there is not enough evidence to confirm whether it benefits hepatic fibrosis [265], and small sample sizes limit current studies. Resveratrol did not improve cardiovascular disease risk factors, such as lipid profiles, serum atherogenic indices, liver enzymes, waist-to-hip ratio, or blood pressure, nor hepatic steatosis [266,267]. Although it is associated with some improvements in inflammatory markers, these changes have not led to clinically significant outcomes for managing metabolic dysfunction-associated steatotic liver disease (MASLD) [268]. 

#### 5.5.6. Choline

Choline is found in certain foods, such as fish and mushrooms, and is also available as a dietary supplement. It resides in membrane phospholipids, which are essential for various biochemical and metabolic processes, including lipid transport, lipid-derived signaling, cholinergic neurotransmission, and the methylation of metabolites. Additionally, it helps maintain the structural integrity of cell membranes [269]. Metabolically, gut microbes break down choline into trimethylamine (TMA), which, upon absorption, is oxidized to trimethylamine-N-oxide (TMAO) by liver enzymes [270]. 

The choline-deficient diets, which reduce TMAO levels in the serum, are linked to MASLD/MASH. In contrast, high serum TMAO levels are associated with CVD and chronic kidney disease. Several studies show that altering the microbial community (microbiome) can cause intestinal dysbiosis through choline-deficient diets, impacting microbiota diversity [270,271]. Therefore, while excessive choline intake may raise TMAO concerns, controlled supplementation could be suitable in specific MASLD phenotypes, especially where choline deficiency is confirmed.

#### 5.5.7. Fish Oil

Fish oil contains omega-3 PUFAs, specifically EPA and DHA. Daily intake of omega-3 PUFAs has been linked to a favorable plasma lipid profile, improved TG, LDL, and ALT levels, and reduced steatosis [272]. One meta-analysis reported a similar decrease in liver fat content and improvement in hepatic enzyme parameters; however, significant heterogeneity was observed between studies [88]. A recent RCT showed a notable reduction in weight and liver fat with 3.6 g/day of n-3 PUFAs [273]. Conversely, another recent RCT found no change in liver fat content, serum AST, ALT levels, or visceral adiposity with daily omega-3 PUFAs supplementation in overweight men [274]. It is important to note that this may be due to higher baseline liver fat levels in other groups [272], compared to this cohort [274]. An ongoing RCT is evaluating the impact of fish oil supplementation on liver fibrosis in patients with metabolic dysfunction-associated liver disease (MASLD) [275].

#### 5.5.8. Co-Enzyme Q10

Co-enzyme Q10 supplementation in MASLD patients showed a non-significant reduction in lipid profile and liver enzymes [276]. However, further research on appropriate dosing, clinical benefits, and adverse effects is needed through large prospective trials involving patients with MASLD.



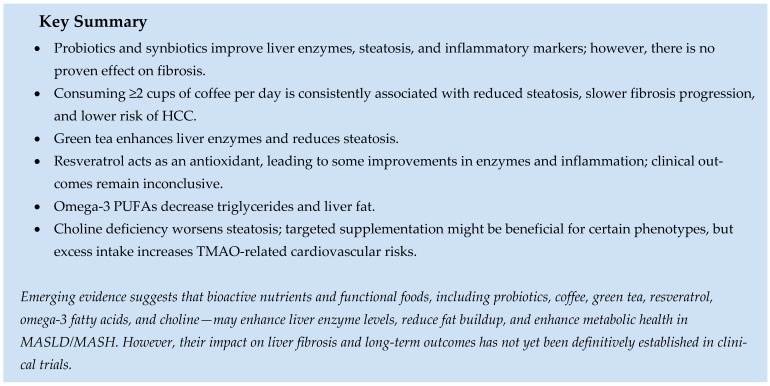



### 5.6. Role of Alcohol, Cannabis, and Tobacco

#### 5.6.1. Alcohol

Regular and heavy alcohol consumption can harm the liver. Cross-sectional studies show that drinking light to moderate amounts of alcohol is linked to benefits in MASLD patients; however, it may also be associated with confounding factors like a low BMI in moderate drinkers [277,278]. In a longitudinal analysis of liver biopsies in MASH patients not on medication, the average alcohol intake improved AST levels. It also decreased the likelihood of MASH resolution compared to non-drinkers [279]. Currently, there is no clear recommendation to use moderate alcohol consumption for MASLD benefits. Alcohol and metabolic dysfunction together may worsen liver disease [280,281,282], and recent data suggest that even light drinking could increase MASLD progression [283]. Another cross-sectional study involving participants with hepatic steatosis from the Framingham Heart Study found a strong link between alcohol use and hepatic steatosis, even after excluding heavy drinkers. This suggests alcohol is a risk factor for MASLD progression [284]. It is also important to consider the type of alcohol consumed, as a study of 1072 participants concluded that those who only drank liquor or cocktails were at higher risk for fibrosis [285]. Any alcohol use, including light drinking, has been independently linked to an increased risk of HCC in MASLD patients [282,286,287]. HCC mainly develops in patients with chronic liver disease. A large study of 4406 reported HCC cases found that MASLD was the underlying disease in 59% of cases [288,289]. A few studies have evaluated the effects of alcohol on the population regarding all-cause mortality. A data review showed that only excessive alcohol use was associated with higher overall mortality compared to non-excessive consumption after an average follow-up of 20 years [290]. A prospective cohort study demonstrated that moderate and heavy alcohol consumption were independently linked to lower and higher all-cause mortality, respectively [291]. Numerous factors complicate the determination of an advisable level of alcohol intake in patients with MASLD, because patients often underreport their alcohol consumption and clinicians find it challenging to estimate lifetime intake. High-quality longitudinal studies are needed to investigate the adverse effects of mild to moderate alcohol consumption and its impact on cardiovascular and liver outcomes in MASLD patients.

#### 5.6.2. Cannabinoids

With increasing legalization and availability, cannabinoid use is rising among various populations. Observational studies have noted a lower prevalence of MASLD among cannabis users [292,293], though therapeutic implications remain uncertain. Cannabinoid receptors—type 1 (CB1) and type 2 (CB2)—are widely present in mammalian tissues, including hepatic and immune cells. CB1 receptors, found in hepatocytes, hepatic stellate cells (HSCs), and liver sinusoidal endothelial cells (LSECs), promote lipogenesis, gluconeogenesis, insulin resistance, and hepatic steatosis when upregulated, as shown in high-fat diet rodent models. CB2 receptors, mainly localized to Kupffer cells (KCs) and HSCs, appear less involved in metabolic injury [294]. In humans, receptor expression patterns in MASLD and cirrhosis suggest a potential role in disease progression, but clear mechanistic links are lacking. A Mendelian randomization study found no causal connection between cannabis use and MASLD risk [295], while cross-sectional analyses of NHANES data (1988–1994 and 2005–2014) indicated that active marijuana use might have a protective effect [293]. Conversely, a large nationwide cohort identified a relationship between cannabis use and a higher prevalence of ascites in individuals with MASLD [296]. These conflicting results, along with limited mechanistic data in humans, highlight the need for more focused research.

#### 5.6.3. Tobacco

Although a direct causal link between tobacco use and MASLD has not been definitively established, evidence suggests a strong association between smoking and disease progression. A large-scale national study found that smoking significantly increases the risk of hepatocellular carcinoma and cardiovascular disease in patients with MASLD [297]. Mechanistically, nicotine has been shown to induce oxidative stress, inflammation, and dysbiosis of the gut microbiota—key pathways involved in the development of MASLD [298]. Additionally, sex-specific cohort analysis indicates that smoking nearly doubles the risk of mortality in MASLD patients, especially in women, highlighting the importance of smoking cessation as a critical intervention in disease management [299]. 



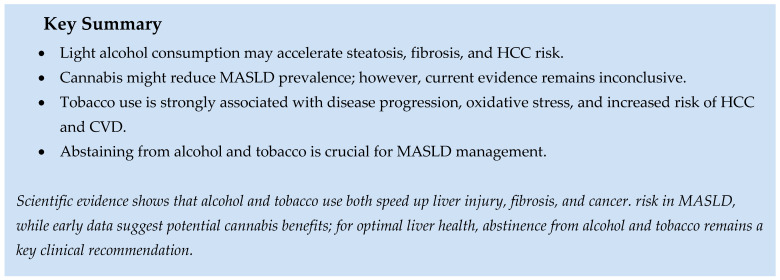



In order to provide a comprehensive perspective, this section ends with a synthesis of all nutrients mentioned above. Table 2 presents these nutrients in a structured format, highlighting their proposed roles in MASLD management. While these herbal supplements and other bioactives have demonstrated improvement of MASLD in small-scale, population-specific trials, large randomized controlled trials are needed to confirm their efficacy and safety in broader MASLD populations.

## 6. Role of Various Dietary Patterns

Dietary patterns define a plan that involves consuming specific macro- and micronutrients while avoiding others. We have discussed the most relevant nutritional practices commonly used to promote weight loss and address various aspects of metabolic syndrome. 

### 6.1. Mediterranean Diet

People in the Mediterranean region eat a diet rich in plant-based foods that are high in antioxidants and anti-inflammatory properties. Mediterranean dietary patterns and lifestyles have been followed for centuries [303]. In traditional MD, a healthy diet includes high MUFA-rich olive oil, moderate intake of nuts, fruits, legumes, vegetables, fish, and wine, and low intake of processed meat, sugar, and dairy products. In MD, about 40% of calories come from fats, mainly MUFAs and omega-3 PUFAs. The MD has a lower omega-6:omega-3 ratio, which is linked to a healthier lipid profile and improved insulin sensitivity [89,304]. Carbohydrate intake in MD provides 40% fewer calories than the 50–60% found in a typical low-fat diet (~60% of calories). Therefore, the EASL-EASD-EASO Clinical Practice Guidelines recommend that the Mediterranean Diet (MD) is a beneficial dietary pattern for patients with MASLD [48]. The MD also lowers the risk of CVD [305] and T2DM [306], and improves overall metabolic health [307,308]. Similar results were supported by an RCT, which demonstrated that the MD was associated with improved LDL cholesterol levels and HbA1c [309].

PREDIMED reports indicated that the overall risk of T2DM can be reduced even without calorie restriction [307]. A detailed meta-analysis showed that the MD benefits the incidence of total CVD and myocardial infarction in individuals with T2DM. Additionally, it revealed that the cohort adhering to the MD exhibited the lowest correlation with overall CVD mortality [310].

A key principle of the MD is to reduce processed, high-sugar foods containing AGEs. AGEs can cause insulin resistance (IR), which may lead to T2DM and other metabolic syndrome conditions, such as high blood pressure and abnormal cholesterol or triglyceride levels [311]. Consistently following the MD also slightly decreases IR, lowers the chance of progressing to advanced liver disease in MASLD patients, and reduces the risk of severe steatosis and steatohepatitis [312]. A prospective randomized trial using nutritional intervention based on the MD found that patients with high adherence to the MD experienced reductions in waist circumference, BMI, and intrahepatic fat content compared to those with low adherence [313]. Studies have shown that the MD is associated with notable improvements in hepatic steatosis [304,314,315,316], insulin sensitivity, liver enzyme profiles, weight loss, and reductions in BMI [304,314].

An RCT assessed the impact of MD (MD group or MDG) or Mediterranean lifestyle (MLG—which includes increased vigorous exercise along with MD) on weight loss in patients with MASLD. Both MDG and MLG showed more significant weight loss compared to the control group. Furthermore, MLG exhibited notable improvements in liver enzyme function and liver stiffness, suggesting that combining additional lifestyle modifications with MD may offer greater overall health benefits for patients with MASLD [317]. Another RCT found that combining MD with an intensive weight loss lifestyle intervention led to greater reductions in triglycerides (TG), fasting glucose levels, and BMI compared to MD alone [318]. In a different RCT, where participants followed both MD and a low-fat diet, results indicated that combining these diets alleviated hepatic steatosis to a similar extent as MDG. However, serum levels of total cholesterol, TGs, HbA1C, and the Framingham risk score were more favorable in the MDG group, reducing all-cause mortality, CVD, type 2 diabetes, cancer, and obesity [314,319]. Researchers have also explored modifications to the MD, such as the green Mediterranean diet (green MD). This variation emphasizes increased intake of green plant-based proteins and polyphenols, found in Mankai, green tea, and walnuts, along with reduced consumption of processed or red meat. The green MD achieved a twice as high intrahepatic fat loss compared to the average MD in an RCT [320].

Overall, MD has strong evidence indicating improvements in insulin resistance (IR), metabolic syndrome, hepatic steatosis, and anthropometric measures such as BMI. It is likely to have a positive influence on cardiovascular risk factors. However, there is no substantial evidence supporting the idea that it improves the histologic features of MASH, which is a key predictor of liver-related mortality. 

### 6.2. Diet Approach to Stop Hypertension Diet

The DASH (Diet Approach to Stop Hypertension) is a diet low in saturated fats, trans fats, and added sugars, but rich in fruits, vegetables, and low-fat dairy products [321]. A case–control study examined the link between following the DASH diet and the risk of metabolic dysfunction-associated liver disease (MASLD). The results indicated that patients who followed the DASH diet had a 30% lower risk of MASLD [322]. Likewise, a cross-sectional study found an inverse association between DASH adherence and MASLD. Participants with the highest compliance to DASH showed a lower risk of MASLD [323]. An RCT conducted among overweight and obese patients with MASLD showed that following the DASH diet had beneficial effects on weight, BMI, liver enzymes, triglycerides, markers of insulin metabolism, and inflammatory markers [324,325]. A calorie-controlled DASH diet results in greater weight loss in overweight and obese adults compared to low-calorie diets. The DASH diet is similar to the Mediterranean (MD) diet. However, there is no evidence that the DASH diet improves fibrosis.

### 6.3. Low-Carbohydrate Diet

A low-carbohydrate diet has been used as a weight-loss strategy since the early 20th century. Such a diet reduces the total daily carbohydrate intake. The low-carbohydrate diet is classified into four groups: (i) Very low-carbohydrate diet (VLCD, less than 10% carbohydrates, or 20–50 g per day), (ii) Low-carbohydrate diet (LCD, 26% carbohydrates or less than 130 g per day), (iii) Moderate-carbohydrate diet (MCD, 44% carbohydrates or 200 g per day), and (iv) High-carbohydrate diet (HCD, more than 44% carbohydrates) [326]. Some VLCDs, like Atkins’ diet, restrict carbohydrate intake to less than 20 g per day while allowing sufficient fat and protein consumption [327]. When carbohydrate intake is below 50 g, ketosis occurs due to glycogenolysis.

The exact mechanism of an LCD remains unclear. However, it is proposed that this approach relies on the carbohydrate-insulin model [328,329,330], which suggests that reducing insulin levels improves cardiometabolic health and aids in weight loss [329]. 

A study evaluated the effects of carbohydrate restriction by comparing a combination of low-fat plus HCD with LCD. An LCD resulted in more weight loss than a conventional diet, but this difference was not significant after one year [331]. An RCT with 609 participants compared the effects of a healthy low-fat diet and a healthy low-carbohydrate diet, finding that the low-carbohydrate group lost more weight than the low-fat group. However, this difference was not statistically significant [332]. Another study showed similar results, indicating that a prolonged hypocaloric LCD had the same effect on reducing intrahepatic lipid accumulation as a low-fat hypocaloric diet [333]. One limitation of this diet is that people on VLCD, high-fat, and high-protein diets (e.g., the Atkins diet) often cannot stick to it for a long time [334].

Low-carbohydrate diets, primarily composed of animal-based protein and fat, such as lamb, beef, pork, and chicken, have been linked to higher mortality rates. In contrast, diets high in plant-based protein and fat, such as nuts, vegetables, and whole grains, are associated with lower mortality, suggesting that the food source significantly influences the relationship between carbohydrate intake and mortality [335].

It is essential to acknowledge the various side effects associated with LCD and VLCD. Several meta-analyses have shown that carbohydrate intake of less than 40% may increase the risk of mortality [335,336]. However, a similar risk has been observed with high carbohydrate intake [337]. According to the Institute of Medicine, Americans are advised to consume 45 to 65% of their calories from carbohydrates [326].

### 6.4. Ketogenic Diet

Humans have evolved to develop metabolic flexibility and the ability to utilize alternative energy sources, such as those beyond exogenous glucose, for optimal function. Instead of relying solely on glucose metabolism, the body activates a metabolic pathway called ketogenesis, which produces ketones from fat in the liver [338]. This pathway, involving the production of ketone bodies, is triggered after glycogen stores are depleted during fasting, low-carbohydrate intake, intense exercise, or starvation. Nutritional ketosis can also be induced by a ketogenic diet, which restricts carbohydrate intake. This diet deliberately increases ketone production, reduces insulin release, and stabilizes blood sugar levels [338]. The goal of this process is to minimize the harmful anabolic effects of insulin. A ketogenic diet typically consists of 5–10% carbohydrates (<20–50 g/day), protein (1–1.5 g/kg/day), and fats until satiety. In the early 1900s, a ketogenic diet was developed to treat seizure disorders [339]. By the 1960s, very low-carbohydrate ketogenic diets (VLCKD) became a standard approach for obesity treatment. The primary purpose of a ketogenic diet is to induce ketosis and stabilize insulin levels, as unstable insulin can disrupt metabolic pathways and lead to insulin resistance [340,341].

The significant changes in lipid biomarkers showed a clear reduction in plasma TG levels and total cholesterol, along with increased HDL levels [339,342]. VLCKD also significantly lowered HbA1C, which indicates overall glycemic control [343,344]. The VLCKD has been effective in achieving long-term reductions in body weight, TGs, and diastolic blood pressure, but was not effective in improving HDL and LDL cholesterol levels compared to a low-fat diet [345].

A study comparing the effectiveness of VLCKD to a standard low-calorie diet found that patients in the VLCKD group experienced greater weight loss, along with significant reductions in visceral adipose tissue and liver fat fraction, compared to those on the standard low-calorie diet [346]. According to another study, a low-carbohydrate, ketogenic diet (hypocaloric) resulted in a more rapid reduction of liver fat and metabolic abnormalities compared to a standard diet [347]. Dysregulated lipid metabolism caused by mitochondrial dysfunction is linked to increased inflammation and oxidative stress, which can lead to hepatocyte death and the progression of MASLD. The ketogenic diet has been shown to reduce oxidative stress and enhance mitochondrial function [348]. A pilot study testing the effects of a low-carbohydrate, ketogenic diet in five patients found it caused significant biopsy-proven histological improvements in MASLD [349]. However, current studies are inconclusive due to small sample sizes and short follow-up periods. Given this limited evidence, it is still difficult to recommend a ketogenic diet as a beneficial treatment for patients with MASLD. 

### 6.5. Low-Fat Diet

Consuming less than 30% or 20% of total daily calories from fat sources is considered a low-fat diet (LFD) or a very low-fat diet (VLFD), respectively [350]. Several studies have compared LCD with LFD to assess their effects on patients with MASLD. In a clinical trial, the Diet Intervention Examining the Factors Interacting with Treatment Success (DIETFITS), found that LCD and LFD were equally effective for weight loss [332]. However, a meta-analysis of 53 randomized clinical trials indicated that the long-term impact of LCD on weight loss was slightly greater than that of LFD [351]. Conversely, a systematic review of 15 clinical trials demonstrated that LFD was highly effective in reducing liver enzymes [352]. Both diets have similar effects on features of metabolic syndromes, such as weight loss and reductions in HbA1C [353]. While this evidence suggests that significant weight loss can be achieved with both LCD and LFD, there is insufficient data to determine which is superior. Additionally, there is insufficient evidence to support their effectiveness in MASLD. A meta-analysis evaluating the effect of LCD (<50% of total energy from carbohydrates) showed that LCD reduced intrahepatic lipid content by approximately 11.52% in patients with MASLD [354]. Ad libitum LFD decreased intrahepatic triglyceride content by 25%, regardless of weight loss in adults [314]. When comparing these diets and their effects on MASLD, data indicate that neither effectively reduces hepatic fat content nor improves aminotransferases in MASLD patients. 

### 6.6. Intermittent Fasting

Intermittent fasting (IF) is a new dietary approach that involves scheduled periods of eating and fasting (calorie restriction), with complete or low-energy intake at regular intervals [355]. According to a systematic review and meta-analysis, intermittent calorie restriction is comparable to continuous calorie restriction for short-term weight loss in overweight and obese adults [356]. A prospective observational study found that the safety and effectiveness of periodic fasting led to a rapid and significant improvement in the fatty liver index (the primary endpoint) in patients with or without type 2 diabetes mellitus (T2DM) [357]. A systematic review and meta-analysis indicated that IF effectively reduces BMI, body weight, fat mass, and total cholesterol in overweight adults [358]. Another systematic review of fourteen studies, along with a meta-analysis of ten studies focusing on individuals with MASLD, showed notable improvements in factors such as body weight, waist-to-hip ratio, BMI, serum AST, serum ALT, and hepatic steatosis following fasting interventions [359]. A randomized controlled trial demonstrated that combining alternate-day fasting with moderate-intensity aerobic exercise resulted in significant reductions in body weight, waist circumference, ALT levels, and increased insulin sensitivity. These improvements were not observed in the exercise-only group, highlighting the role of intermittent fasting in achieving these benefits [360]. IF triggers a metabolic pathway that shifts lipid and cholesterol synthesis and fat storage mechanisms toward fat mobilization through fatty acid oxidation and ketone production. These processes help optimize physiological functions, slow aging, and prevent disease progression [361]. A major challenge with IF is maintaining long-term adherence to this dietary pattern. A clinical study evaluated the effectiveness and adherence of 8 weeks of modified alternate-day calorie restriction (MACR). In control subjects, implementing MACR showed lower MASLD activity compared to usual diets, with reasonable adherence rates [362]. The positive effects of IF are significantly enhanced when combined with aerobic exercise, leading to reductions in body weight, fat content, waist circumference, ALT, and an increase in insulin sensitivity [360]. By promoting weight loss, IF can potentially have a beneficial impact on MASLD.



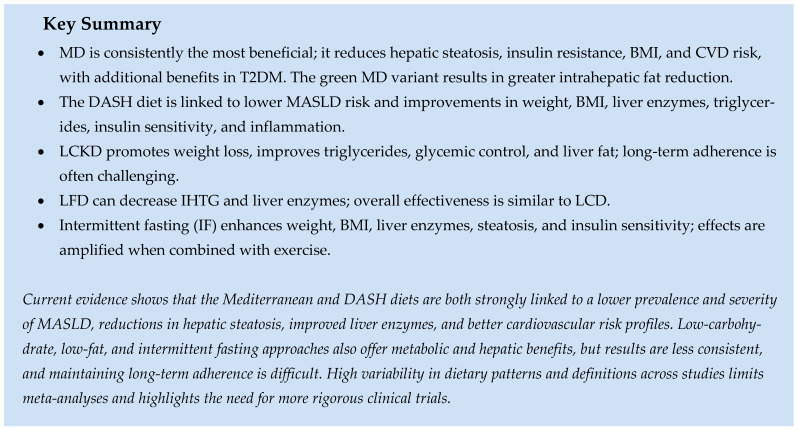



## 7. Role of Physical Activity and Exercise

Regular physical activity is a fundamental part of MASLD management. Physical inactivity is an independent predictor of MASLD, whereas exercise provides hepatic and metabolic benefits beyond just weight loss [363]. A recent cross-sectional NHANES study (2017–2018) found a prevalence of sarcopenia of 11.7% in MASLD patients compared to 3.8% in those without MASLD. Importantly, individuals with sarcopenia who also had lower physical activity levels faced a 7.91-fold higher risk of significant fibrosis, highlighting exercise’s role in reducing muscle loss and fibrosis risk [364]. Sarcopenia, common in MASLD, worsens disease progression and increases fibrosis risk. The importance of skeletal muscle health is increasingly acknowledged. Beyond these effects, exercise exerts multi-system benefits, which are detailed below and summarized in Figure 2.

Exercise reduces liver fat and improves metabolic markers through various mechanisms, such as enhanced insulin sensitivity, adipose tissue remodeling, better muscle metabolism, altered hepatokine signaling, improved mitochondrial function, and decreased inflammation. Aerobic, resistance, or combined exercise types consistently improve hepatic steatosis and cardiometabolic health, supporting guidelines that recommend 150–300 min of moderate-intensity activity or 75–150 min of vigorous activity weekly. Resistance training is particularly helpful for fighting sarcopenia, frailty, and age-related muscle loss common in advanced disease [365].

Large cohort and prospective studies confirm that exercising more than five times a week significantly reduces the risk of hepatic steatosis and promotes resolution, independent of weight loss [366]. Physical activity shows an inverse, dose-dependent relationship with MASLD severity, cardiovascular risk, and mortality [367,368,369,370,371,372]. Emerging evidence suggests that even concentrated exercise patterns, such as the ‘Weekend Warrior’ approach involving one or two sessions per week, provide similar protection against MASLD and mortality [373]. Conversely, sedentary behavior is a strong risk factor for MASLD and all-cause mortality [369,374,375,376,377].

Mechanistic studies indicate that exercise reduces hepatic inflammation by inhibiting the MD 2-TLR 4 pathway and slows MASH progression by decreasing hepatic monocyte-derived inflammatory macrophages and bone marrow precursor cells [378,379,380]. Systemic effects include reduced adipocyte size, increased adiponectin levels, improved skeletal muscle mitochondrial function and glucose uptake, enhanced secretion of beneficial myokines (e.g., irisin), and better gut microbiota diversity and barrier integrity. These systemic adaptations highlight why exercise is both therapeutic and preventive across the MASLD spectrum, reinforcing the need to incorporate it into daily routines.

However, reports indicate that most patients with MASLD do not meet these exercise recommendations [363,381], engaging in less moderate to vigorous exercise than their healthy counterparts [382]. Exercise intensity significantly influences the reduction of CVD and metabolic disease risk [383]. Vigorous activities can substantially lower CVD risk and decrease overall body fat, visceral fat, and blood pressure in MASLD patients [370,371,372]. Physical activity has been shown to significantly reduce postprandial triglyceride levels and improve post-meal fat oxidation [384]. Patients are encouraged to create a structured exercise routine and stick to it strictly to fight MASLD. The FITT Principle (Frequency, Intensity, Type, Time) offers a practical framework to tailor individual exercise plans and enhance adherence. Supervision and structured support improve outcomes, although excessive intensity may lead to higher dropout rates.

Aerobic and resistance exercises lower the risk of hepatic steatosis in patients with MASLD. Combining both types of exercise is the most effective way to improve weight, waist circumference, triglycerides, cholesterol, glucose, and insulin levels [385]. However, resistance exercises have significantly lower exercise intensity and energy consumption [386]. Therefore, they are recommended for patients who are unable to participate in vigorous aerobic exercises. While both exercise types offer benefits, it is wise to encourage patients to start with any form they can consistently do on a daily basis.

### 7.1. Aerobic Exercise 

Aerobic exercise (AE), defined as activity that requires increased oxygen consumption compared to rest, improves blood flow, reduces hepatic steatosis, and enhances metabolic health in MASLD [381]. Clinical studies demonstrate that AE can decrease fatty acid synthesis and liver fat content by 2–50%, thereby helping to alleviate hepatic steatosis [387,388,389], a finding that is notably independent of weight loss [390]. These benefits extend to modest reductions in aminotransferases [388], improved insulin resistance (IR), better lipid profiles, and help preserve muscle mass [391]. A meta-analysis of 11 randomized controlled trials (RCTs) evaluating AE’s effects on MASLD showed a significant reduction in triglycerides (TGs), low-density lipoprotein (LDL), and an increase in high-density lipoprotein (HDL). It also found reductions in AST and ALT levels [392]. A clinical trial found that 12 weeks of aerobic training reduced fibrosis and hepatocyte ballooning by one stage in 58% of patients [393]. Even relatively modest regimens, such as 135 min per week of moderate aerobic activity, have been shown to reduce hepatic fat in MASLD.

At the cellular level, AE has been shown to decrease oxidative stress and reduce inflammation by releasing anti-inflammatory cytokines [394]. It enhances cardiorespiratory fitness, thereby lowering CVD risk and benefiting MASLD patients, who experience lower mortality [395]. Moderate to vigorous exercise for 150 min per week has been studied in the general population to evaluate its benefits in reducing hepatic steatosis or alleviating steatosis. It was found to be effective regardless of weight loss in patients with MASLD [371]. 

### 7.2. High-Intensity Interval Training

High-Intensity Interval Training (HIIT) is a structured form of aerobic exercise that involves short bursts of high-intensity activity followed by brief recovery periods to increase calorie burning. This method aims to fit intense workouts into a short amount of time [396,397]. While HIIT can be challenging to perform, it allows for rapid calorie burning, muscle building, and weight loss. It also enhances cardiorespiratory fitness in a short period more effectively than other exercises performed for longer durations [398]. This intense exercise, characterized by short spurts, has been shown to improve hepatic steatosis and cardiac function in MASLD patients [39]. However, high-intensity exercises do not significantly reduce hepatic steatosis compared to moderate-intensity exercises [370,399].

However, vigorous-intensity exercises substantially boost aerobic fitness [400] and decrease CVD risk in patients with MASLD [370,372]. It has also been shown to lead to a greater reduction in postprandial TG and an increase in postprandial fat oxidation, which reduces the risk of CVD and insulin resistance [384,401].

### 7.3. Resistance Exercise

Resistance exercise (RE), which involves repeated muscle contraction against a load, enhances muscle strength, metabolic function, and skeletal health. RE modestly reduces hepatic steatosis [402], improves glycemic control and HbA1C in patients with T2DM [403], and increases insulin sensitivity by promoting GLUT-4 expression in skeletal muscles [404]. It helps delay skeletal muscle loss [405], boosts lean mass [406], decreases intramuscular lipid buildup [407,408], and improves microvascular blood flow in skeletal muscles [409].

RE can serve as an alternative exercise option for patients with limited cardiopulmonary capacity who cannot participate in moderate to vigorous aerobic exercises [410]. RE results in greater post-exercise metabolic activity and energy expenditure [411]. A systematic review and meta-analysis found that resistance exercises were more effective in reducing hepatic fat content, body fat, and metabolic syndrome, along with cardiovascular risk factors. Conversely, aerobic exercise was more effective for BMI reduction [411].

In clinical practice, combining AE and RE provides complementary benefits, leading to optimal improvements in hepatic and metabolic outcomes. Depending on their age, medical co-morbidities, and cardiorespiratory capacity, they should also evaluate which type of exercise best suits the patient’s individual needs. Physical activity and exercise are most effective for MASLD patients when a personalized and tailored approach is used, making RE essential for maintaining lean mass and reducing frailty in sarcopenic or older MASLD patients. Figure 3 shows the physiological effects of different workout types, while a subgroup analysis of exercise data in MASLD patients is presented in Table 3.



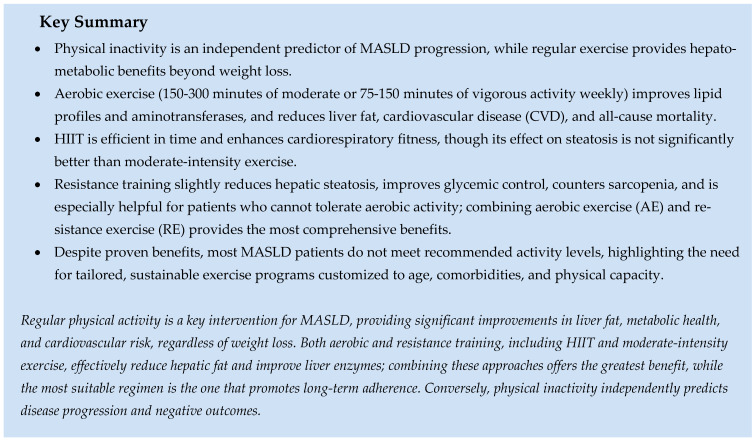



## 8. Role of Circadian Rhythm in MASLD

Circadian rhythm disruption is increasingly recognized as a key factor in the development of MASLD. When circadian cycles are misaligned, it disrupts neuroendocrine and metabolic regulation, leading to liver damage. Zhang et al. demonstrated that social jetlag, a form of circadian misalignment, disrupts prolactin rhythms and promotes the expression of lipogenic genes, resulting in hepatic steatosis. Their research also emphasized the importance of chronopharmacology, showing that the effectiveness of drugs in MASLD depends on the timing of administration [420]. Animal studies further indicated that circadian disturbances could occur before visible liver changes, suggesting they could serve as early diagnostic markers [421]. Human studies, including meta-analyses, support these findings by linking poor sleep hygiene and circadian misalignment to increased risk of metabolic disorders through changes in eating habits and gut microbiota imbalances [422]. Clinical data from Schaeffer et al. show that MASLD patients often experience fragmented sleep, with extended wakefulness during the night and reduced sleep efficiency, which worsens features of metabolic syndrome [423]. Additionally, Jain et al. found that diet-induced obesity alters mitochondrial composition, further disrupting circadian regulation and exacerbating steatosis [424]. Altogether, these findings highlight a reciprocal relationship between circadian rhythm and metabolic balance, making circadian alignment a promising target for the prevention and treatment of MASLD. 

## 9. Psychosocial Determinants

Psychological comorbidities are a vital yet often neglected factor that influences adherence. Depression affects about 18% of individuals with MASLD and is independently associated with lower quality of life, unhealthy lifestyle choices, and more severe histological features, including ballooning and fibrosis, especially in MASH. Anxiety is also common, with a pooled rate near 37%, and may contribute to poorer quality of life and more advanced histological problems, although these links differ by sex and disease stage.

Stress, although less extensively studied, has been linked to metabolic dysfunction and may independently raise the prevalence of MASLD. However, available data is limited and varied, highlighting the need for further research [425].

These findings emphasize that effective lifestyle interventions in MASLD must encompass not just diet and exercise but also address psychological and social obstacles to consistent treatment. Screening for depression, anxiety, and stress, combined with providing behavioral and psychological support, is essential for improving adherence and sustaining long-term therapeutic benefits.

## 10. Barriers to Implementing Lifestyle Interventions

Changing routine dietary and lifestyle habits can be very challenging because it involves recognizing what needs to change, making those adjustments, and maintaining the new behaviors over time. The first challenge in understanding MASLD is diagnosing it and grasping the negative effects it has on health. Patients often say they receive insufficient information and support about their diagnosis [426]. Therefore, using an empathetic approach and simplifying complex information about the diagnosis and its clinical effects can help improve understanding of the disease and its impact. Next, identifying what needs to change—such as eating habits, inactivity, or substance use—is essential to achieving the desired outcomes. Behavior change techniques are useful for guiding patients in deciding which aspects of their behavior and lifestyle should be modified [427,428,429]. This may involve reviewing behavioral goals, identifying barriers, and using follow-up prompts to track progress [427]. Encouraging patients to self-monitor their daily activities and fitness routines with pedometers, smartphone fitness apps, or by keeping an activity journal has proven effective in supporting long-term adherence and consistency in physical activity and lifestyle changes [429].

## 11. Future Directions

The future of MASLD management lies in precision lifestyle medicine, where diet, exercise, and technology work together to provide personalized care. While diet remains the foundation, important questions still exist: which approach—Mediterranean, plant-based, or intermittent fasting—offers the greatest benefit for specific patient groups? Nutrient-specific effects are equally important. Fructose, saturated fats, trans fats, and alcohol accelerate disease progression, whereas fiber, polyphenols, resistant starches, and omega-3 fatty acids provide protection. Many of these effects are mediated through microbiome-derived metabolites that regulate fat storage, inflammation, and fibrosis. Future research should compare nutrient-specific strategies, like fructose restriction, with broader dietary patterns to determine which offers the most sustained hepatic and cardiometabolic benefits.

Skeletal muscle health is becoming an increasingly important focus. Myosteatosis worsens insulin resistance, promotes steatosis, and heightens cardiovascular risk. Resistance training, especially when combined with aerobic exercise, can counteract these effects by enhancing mitochondrial function, increasing muscle mass, and improving glucose disposal. Understanding the best type, timing, and intensity of exercise, along with how muscle signals (myokines) influence the liver, will be essential. Targeting muscle–liver communication through structured exercise may be as crucial as diet in MASLD management. Additionally, recognizing sex-specific, age-related, and sarcopenia-related differences will help refine exercise recommendations.

Beyond muscle, inter-organ communication involving adipose tissue, the liver, and the gut microbiota offers new therapeutic possibilities. Microbiome profiling might allow for personalized use of probiotics, symbiotics, or targeted nutrition. Pharmacologic innovations, such as GLP-1 receptor agonists, FGF21 analogs, mitochondrial enhancers, and agents that promote adipose browning, are likely to work best when combined with lifestyle strategies. Multi-omics approaches, integrating genomics, epigenetics, microbiome, and metabolomics, can help identify which patients will respond and stratify them effectively. Hybrid trial designs that combine lifestyle, pharmacological, and microbiome-based interventions could speed up the clinical application of these methods.

Technology will play a vital role in this development. Artificial intelligence can generate personalized diet and exercise plans, predict individual responses, and improve adherence through adaptive feedback. AI-driven multi-omics integration might enable “custom diets” tailored to each person’s genomic, metabolic, and microbial profiles. Wearable devices, continuous glucose monitoring, and mobile health apps can further support self-care, especially in resource-limited or remote areas. However, issues like data privacy, algorithm bias, and equitable access need to be addressed.

Future research should also examine factors beyond diet and exercise. Sleep quality, circadian rhythm, and early-life exposures greatly influence metabolic health. Preventive strategies for children, adolescents, and even during prenatal stages provide opportunities to reduce lifetime disease risk. At the community level, tackling food insecurity, environmental hazards, and health disparities is essential. Policy initiatives, community programs, and behavioral interventions, such as motivational interviewing and family-based approaches, can foster lasting lifestyle changes.

Ultimately, MASLD research must go beyond histological endpoints. Long-term outcomes, such as cardiometabolic risk reduction, the sustainability of interventions, and equitable access, should become primary goals. By integrating dietary optimization, exercise science, microbiome research, pharmacological innovation, and AI-driven personalization within a socioecological framework, MASLD care can move from a “one-size-fits-all” approach to precise, scalable, and equitable solutions. Figure 4 illustrates how lifestyle medicine can be integrated to achieve precision in MASLD. 

## 12. Conclusions

Diet plays a crucial role in managing MASLD. Consuming excess calories, especially from ultra-processed foods high in refined sugars and saturated fats, promotes de novo lipogenesis, visceral fat accumulation, and chronic inflammation, all of which lead to persistent insulin resistance. Conversely, diets centered on whole foods, such as Mediterranean-style eating patterns rich in fiber, polyphenols, and monounsaturated fats, improve insulin sensitivity by reducing oxidative stress and promoting metabolic balance. Importantly, food quality is just as vital as quantity, since additives and lower antioxidant intake can further interfere with insulin signaling, and alcohol consumption can worsen liver inflammation in these conditions.

Physical activity enhances the benefits of dietary efforts by increasing glucose uptake, improving mitochondrial function, and strengthening the interaction between muscle and liver metabolism—ultimately reducing insulin resistance. Aerobic exercise decreases liver fat and systemic inflammation, providing cardiovascular benefits even without significant weight loss. Resistance training helps maintain and rebuild lean muscle mass, improves insulin-mediated glucose metabolism, and fights sarcopenia and muscle fat infiltration. Combining aerobic and resistance exercises yields even greater improvements in metabolic flexibility. Despite these benefits, maintaining lifestyle changes remains challenging. New digital health tools—such as AI-driven meal planning, wearable devices, and behavior-focused apps—offer personalized, scalable support to help follow targeted interventions for insulin resistance.

Future MASLD treatments should go beyond generic advice. Strategies need to be tailored based on individual metabolic profiles, comorbidities such as diabetes, dyslipidemia, sarcopenia, and genetic factors, especially PNPLA3 and TM6SF2 variants that affect nutrient processing, fat storage, and antioxidant needs. Specific clinical frameworks are also crucial for cases involving lean-MASLD and atypical fat distribution. Current research faces limitations due to short study durations, lack of patient stratification, and reliance on surrogate markers. Long-term trials examining macronutrient ratios, antioxidant supplements, and combined exercise programs are essential to understand their impact on disease remission, fibrosis progression, and cardiometabolic health outcomes. Additionally, genotype-informed lifestyle interventions must be validated in populations at high risk of the disease.

In summary, optimized MASLD care requires a precision medicine approach that targets insulin resistance in the liver, fat tissue, and skeletal muscle. Improving dietary quality, customizing exercise routines, and adding behavioral support with genetic and digital tools are essential for reversing the underlying pro-inflammatory and insulin-resistant processes that drive disease progression. Using mechanistic knowledge to develop sustainable, personalized therapies will be crucial for future success.

## Figures and Tables

**Figure 1 ijms-26-09625-f001:**
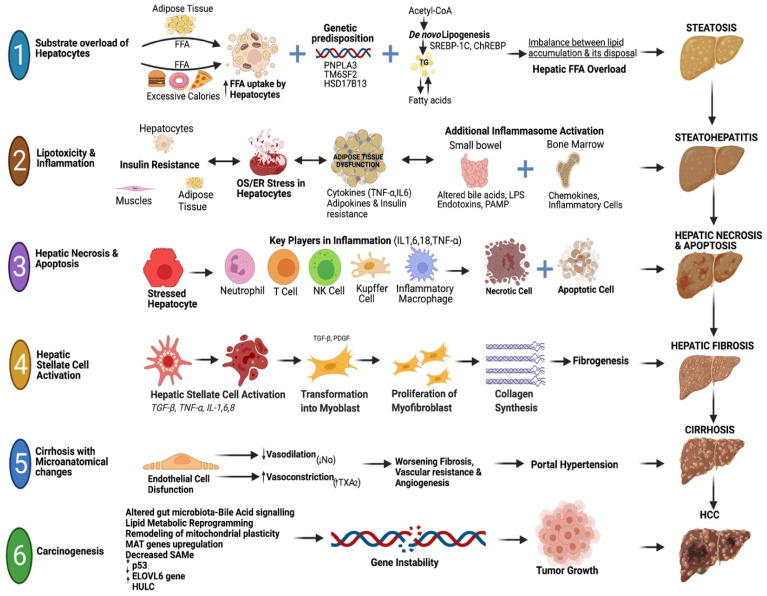
Disease progression involves multiple parallel “hits”: 1. Substrate Overload of Hepatocytes: Excess calories, metabolic syndrome, and genetic variants (PNPLA3, TM6SF2, HSD17B13) promote steatosis through increased FFA uptake (CD36, FATP2/5, PPARδ, FABP1) and de novo lipogenesis (SREBP1c, ChREBP). Impaired β-oxidation (e.g., hypothyroidism) worsens lipid accumulation. 2. Lipotoxicity and Inflammation: Elevated FFAs contribute to insulin resistance (muscle, adipose, liver), oxidative stress, cytokine release, and dysbiosis-related endotoxin exposure, fueling hepatocellular inflammation. 3. Hepatic Necrosis and Apoptosis: Lipid toxicity and cytokines (TNF-α, IL-1β, IL-6, IL-18, IL-33, MCP-1) induce apoptosis/necroptosis, recruiting neutrophils, macrophages, Kupffer cells, T/NK cells, and activating stellate cells. 4. Hepatic Stellate Cell Activation: Injury and immune signaling (TGF-β, PDGF, LPS) transform HSCs into collagen-producing myofibroblasts, leading to fibrosis. 5. Cirrhosis with Microanatomical Changes: Lipid accumulation and hepatocyte ballooning disrupt LSECs, elevate portal pressure, and promote endothelial dysfunction (↓NO, ↑TXA2). 6. Carcinogenesis: Advanced fibrosis increases the risk for HCC through altered microbiome-bile acid signaling, mitochondrial and lipid reprogramming, MAT/SAMe imbalance, p53 loss, HULC upregulation, and miRNA dysregulation. Abbreviations: CD: cluster of differentiation; ChREBP: carbohydrate regulatory element-binding protein; ER: endoplasmic reticulum; EV: extracellular vesicles; FABP1: fatty acid binding protein I; FATP: fatty acid transport protein; FFA: free fatty acids; GCKR: glucokinase regulator; HCC: hepatocellular carcinoma; HILC: highly upregulated in liver cancer; HSC: hepatic stellate cells; HSD17B13: 17b-Hydroxysteroid dehydrogenase type 13; HULC: human universal-length non-coding RNA; ILC: innate lymphoid cells; IL: interleukin; MBOAT7: membrane-bound O-acyl transferase domain-containing 7; KCs: Kupffer cells; LPS: lipopolysaccharide; OS: oxidative stress; LSECs: liver endothelial sinusoidal cells; MAT: methionine adenosyl transferase; miRNA: micro ribonucleic acid; NO: nitric oxide; PAMP: pathogen-associated molecular pattern; PDGF: platelet-derived growth factor; PNPLA3: patatin-like phospholipase domain-containing 3; PPAR: peroxisome proliferator-activated receptor gamma; SAMe: S-adenosylmethionine; SREBP1c: sterol regulatory element-binding protein 1c; VLDL: very low density lipoprotein; TGF: transforming growth factor; TNF: tumor necrosis factor; Th: T helper; TXA2: thromboxane A2.

**Figure 2 ijms-26-09625-f002:**
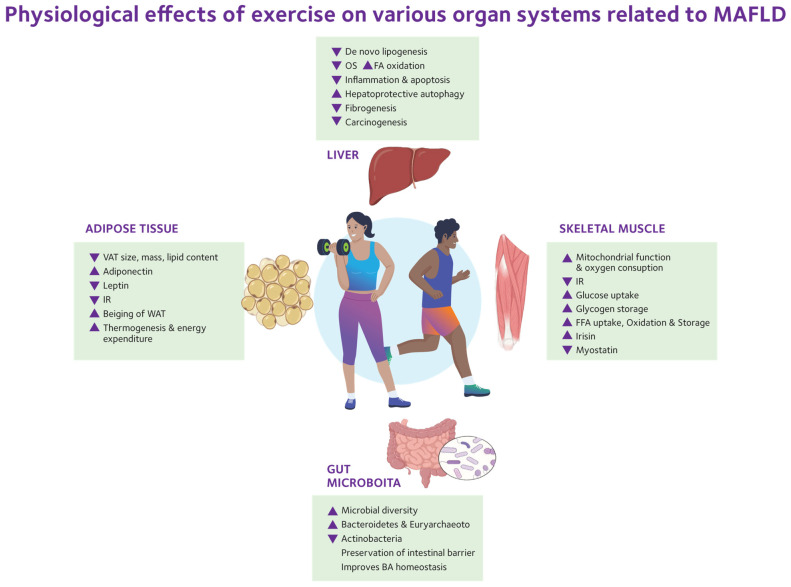
Adipose Tissues: A sedentary lifestyle and a high-calorie diet cause adipocyte enlargement, which induces oxidative and ER stress, increases lipolysis, and leads to the spillage of FFA into the circulation, resulting in cell death and inflammation in adipose tissue. Exercise, especially aerobic exercise, reduces adipocyte size and mass, increases adiponectin, and decreases leptin, thereby lowering systemic inflammation associated with metabolism. Exercise also promotes the initiation of WAT and elevates protein uncoupling, leading to thermogenesis and greater energy expenditure. Skeletal Muscles: Exercise enhances capillary density, mitochondrial function, and oxygen consumption capacity in skeletal muscles by upregulating PPAR-γ and PGC1-α. It increases glucose uptake via the GLUT-4 transporter, improving insulin sensitivity and boosting fatty acid uptake, oxidation, and storage. Additionally, exercise enhances glycogen storage in muscles and influences myokines such as irisin and myostatin. Irisin release is stimulated by exercise, promoting WAT formation and upregulating PPAR-γ and FGF-21, which exert a direct anti-steatogenic effect on the liver. Conversely, exercise reduces myostatin release, which has pro-fibrogenic effects through direct action on HSCs. Liver: Exercise decreases SREBP-1c levels, reducing de novo lipogenesis, while increasing PPAR-γ expression, which enhances hepatic mitochondrial fatty acid oxidation and reduces oxidative stress by activating antioxidant defenses and increasing enzyme activity, such as catalase, superoxide dismutase, and glutathione peroxidase. Exercise also exerts anti-inflammatory effects on the liver by inhibiting the release of pro-inflammatory cytokines. It promotes hepatoprotective autophagy and slows the progression of steatosis to steatohepatitis, fibrosis, and cancer. Gut Microbiota: Exercise diversifies gut microbiota by balancing the growth of Bacteroidetes, Euryarchaeota, and Actinobacteria. It also helps preserve the intestinal barrier and maintain bile acid homeostasis. Abbreviations: BA: bile acid; ER: endoplasmic reticulum stress; FFA: free fatty acids; FGF-21: fibroblast growth factor 21; GLUT-4: glucose transporter type 4; HSC: hepatic stellate cells; IR: insulin resistance; OS: oxidative stress; PGC1-α: peroxisome proliferator-activated receptor gamma coactivator 1 alpha; PPAR-γ: peroxisome proliferator-activated receptor gamma; SREBP-1c: sterol regulatory element-binding protein 1c; VAT: visceral adipose tissue; WAT: white adipose tissue.

**Figure 3 ijms-26-09625-f003:**
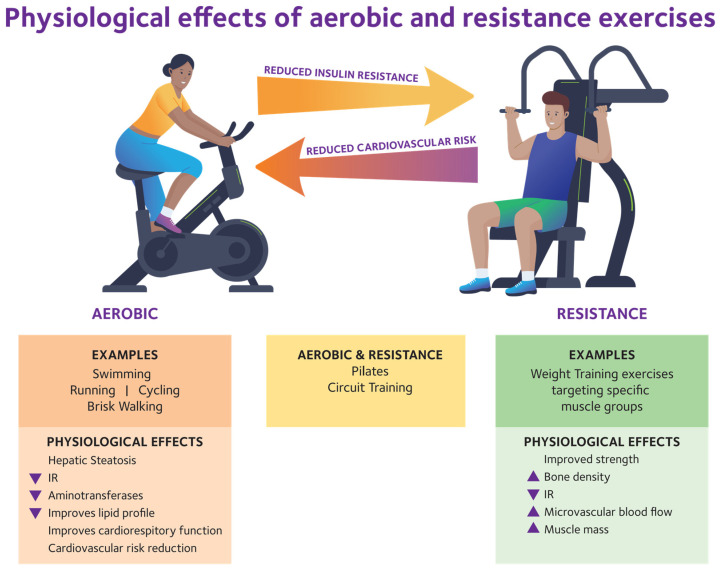
Physiological effects of aerobic and resistance exercises: examples.

**Figure 4 ijms-26-09625-f004:**
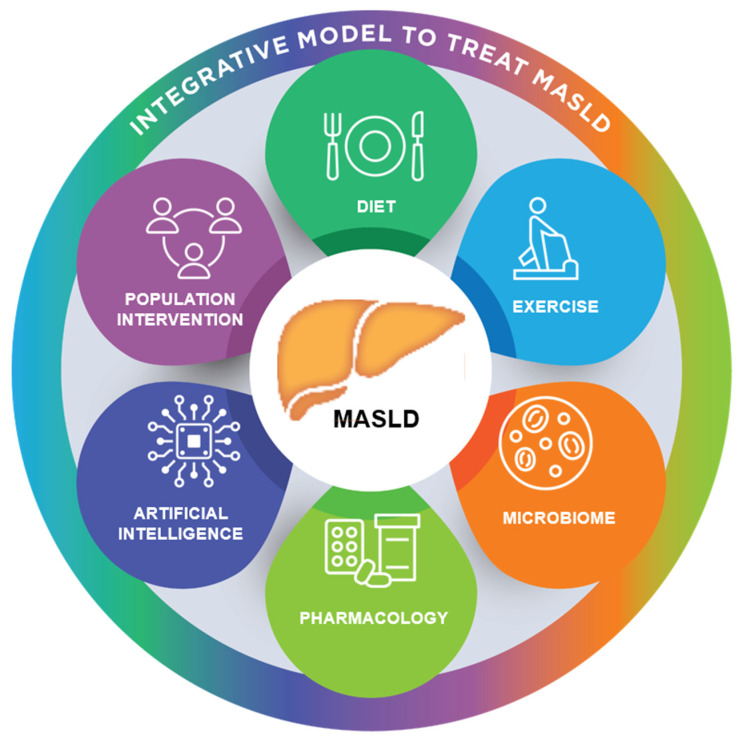
Precision lifestyle medicine in MASLD. Diet, exercise, and technology form the foundation, supported by sleep and social factors. Inter-organ communication and multi-omics profiling facilitate patient stratification and the development of targeted treatments.

**Table 1 ijms-26-09625-t001:** Guidelines for weight loss in MASLD in various societies.

Society Guidelines	% Weight Loss	Dietary Restrictions	Physical Activity	Ref.
**AASLD (2023)**	3–5% weight loss improves steatosis. 7–10% weight loss improves most histopathologic features of MASH, including fibrosis.	A diet that leads to a caloric deficit and is limited in carbohydrates and saturated fats. Mediterranean dietary pattern.	Aerobic exercise at least five times a week for a total of 150 min/week	[47]
**EASL-EASD-EASO (2024)**	3–5% weight loss for MASLD with average weight ≥5% weight loss for steatosis reduction ≥7–10% weight loss for MASH and fibrosis reduction	Recommend the Mediterranean diet. Minimizing processed and ultra-processed foods while increasing the intake of unprocessed or minimally processed foods.	>150 min/week of moderate or 75 min of vigorous exercise. Minimizing sedentary time.	[48]
**KASL (2021)**	7–10% weight loss.	Calorie restriction (500 kcal), low-carbohydrate, and low fructose diet.	Exercising for at least 30 min. Three times a week	[49]
**APWP (2025)**	>5% for steatosis reduction 7–10% for MASH resolution >10% improves liver fibrosis	1200–1800 kcal/day or 500–750 kcal caloric restriction Low-carbohydrate and ultra-processed food abstinence Mediterranean diet, ketogenic diet, intermittent fasting, and time-restricted feeding are recommended	150–240 min/week of moderate-to-vigorous intensity aerobic exercise 2–3 days/week of resistance training	[50]
**NICE (2016)**	Consider NICE guidelines for obesity and weight gain prevention.	Consider NICE guidelines for obesity and weight gain prevention. No specific diet.	Consider NICE guidelines for obesity and weight gain prevention.	[51]
**ADA (2025)**	≥5% decreases steatosis ≥10% improves fibrosis	Mediterranean diet benefits on cardiometabolic factors; highly saturated fats, carbohydrates, and alcohol should be avoided	150 min/week of moderate or 75 min/week of rigorous aerobic 2–3 times/week of resistance training	[8]

**Abbreviations:** AASLD: American Association for the Study of Liver Diseases; Asia-Pacific Working Party on Non-Alcoholic Fatty Liver Disease; EASL-EASD-EASO: European Association for the Study of the Liver/European Association for the Study of Diabetes/European Association for the Study of Obesity; KASL: Korean Association for the Study of the Liver; APWP: Asia-Pacific Working Party; NICE: National Institute for Health and Care Excellence; ADA: American Diabetes Association.

**Table 2 ijms-26-09625-t002:** Summary of currently available nutritional data in MASLD. Effects of micronutrients, minerals, herbal supplements, and other components in human (H) and animal (A) studies.

Summary of Current Nutritional Data in MASLD H = Human, A = Animal		
**Calories**	Daily restriction of 500–1000 kcal results in an improvement in insulin resistance and hepatic steatosis (H)	[9,48]
**MACRONUTRIENTS**		
**Fats**	SFAs (found in dairy products, vegetable oils, desserts, and red meat) increase intrahepatic triglycerides and plasma ceramides, impairing insulin sensitivity (H). RCTs show increased liver fat and ceramides with high SFA diets, with a strong link to fibrosis progression.	[72,73,74,75,76,77,78]
	Increased intake of MUFAs (found in olive oil, avocados, and nuts) is associated with a healthier lipid profile (lower LDL cholesterol, triglycerides, and a reduced total cholesterol/HDL ratio), decreased lipotoxicity, and improved insulin sensitivity (H). Olive oil, a key component of MD, has antioxidant, anti-inflammatory, and antithrombotic properties, which help improve steatosis and reduce cardiovascular risk (H).	[79,80,81,82,83,84,85]
	Omega-3 and -6 PUFAs are essential fatty acids obtained solely through diet. Increasing intake of Omega-3 PUFAs (found in chia and flax seeds, walnuts, salmon, and dietary supplements) lowers hepatic triglyceride levels, reduces hepatic steatosis, and enhances insulin sensitivity (H). They have anti-inflammatory and anti-fibrotic properties.	[86,87,88,89,90,91,92,93,94,95]
	Increased intake of Omega-6 PUFAs (found in vegetable oils) is linked to a higher risk of CVD, cancer, inflammation, and autoimmune diseases. Omega-6 PUFAs are associated with inflammation if their ratio to Omega-3 is high.	[86,87,88,89,90,91,92,93,94,95]
	Increased intake of trans-fats (found in baked and refrigerated foods) has a pro-oxidative effect, leading to increased insulin resistance, obesity, and systemic inflammation, and is associated with an increased risk of developing MASLD in animal studies (A). Human evidence is limited. Clinical Guidance supports strict avoidance.	[68,96,97,98,99,100]
**Carbohydrates**	Carbohydrates are the most abundant macronutrients and can be classified as simple or complex. The dietary source of carbohydrates plays a crucial role in determining its effect on patients with MASLD. Simple carbohydrates (found in sugar-sweetened beverages) pose a high-risk factor for MASLD patients (H).	[67,102]
	Refined and added carbohydrates lead to an increase in glycemic load, causing hyperinsulinemia, insulin resistance, increased DNL, visceral adiposity, and hepatic fat. Observational and interventional studies show strong links with MASLD prevalence and progression.	[102,103,104,105,106,107,113,114,115,116,117,118]
	RCTs and cohort studies have linked fructose intake to steatosis, MASH, and fibrosis progression, and higher serum fructose levels have been correlated with MASLD risk.	[108,109,110,111,112,113,114,115,116,117,118]
	Lack of dietary fiber (a type of carbohydrate) in the diet has been linked to MASLD. Prebiotic fibers and non-digestible carbohydrates (e.g., resistant starch) modulate gut microbiota and significantly improve serum AST, ALT, insulin, and IHTG levels, while also reducing inflammation (H). A protective association has been observed between reduced steatosis and metabolic risk in cohort and dietary intervention studies.	[105,106,107,119,120,127]
**Proteins**	Excessive consumption of red meat, especially processed meat, raises the risk of MASLD, T2DM, CVD, and death in patients by fostering insulin resistance (H).	[129,130,131,300,301,302]
	Processed meat is strongly linked to MASLD and all-cause mortality. It is high in sodium, nitrates, and preservatives, and it worsens metabolic and inflammatory pathways.	[129,130,300,301,302]
	Fish, eggs, and plant-based proteins provide high-quality protein, along with omega-3 fatty acids and choline. This leads to decreased steatosis, as well as anti-inflammatory and antifibrotic effects.	[105,128,300,301]
**MICRONUTRIENTS**		
**Vitamin E**	Daily supplementation of Vitamin E (800 IU) in non-diabetic patients improved histologic features of MASH (H).	[47,48,144]
**Vitamin C**	Daily supplementation with a combination of Vitamin C and E (1000 mg and 1000 IU, respectively) is inversely related to the severity of MASLD and shows improvement in fibrosis scores (H).	[152]
**Vitamin D**	Vitamin D supplementation may exert antifibrotic, anti-inflammatory effects (H). Vitamin D deficiency is associated with increased IR and may predispose to MASLD (H).	[156,160,161]
**Vitamin A**	Vitamin A deficiency in patients with MASLD may be associated with the progression of MASLD (H).	[169]
**Vitamin B3**	Vitamin B3 reduced IHTG (H). Niacin treatment showed improvement in TGs, VLDL, and insulin sensitivity (H).	[174,179]
**Vitamin B6**	Vitamin B6 supplementation (90 mg daily) significantly ameliorated hepatic fat accumulation (HFA).	[182]
**Vitamin B9**	Vitamin B9 deficiency was considered an independent risk factor in MASLD (A). Folate supplementation ameliorates hepatic steatosis and reduces pro-inflammatory cytokines (A).	[187,188]
**Vitamin B12**	Low levels of vitamin B12 are associated with increased severity of MASH (H).	[188]
**Calcium and Phosphorus**	High serum calcium and phosphorus levels may be associated with MASLD (H).	[194]
**Magnesium**	A high intake of magnesium may be associated with a reduced risk of MASLD (H).	[195]
**Zinc and Selenium**	In animal studies, zinc and selenium supplementation improved serum AST, ALT, triglycerides, and total cholesterol in MASLD (A).	[196]
**Iron**	Iron was associated with worsening steatohepatitis in animal models (A).	[197]
**HERBAL SUPPLEMENTS**		
**Milk Thistle**	Silymarin (milk thistle plant extract) has been shown to have antioxidant, anti-inflammatory, and antifibrotic effects (H). It reduces oxidative damage, hepatic steatosis, and IR in MASLD (H).	[199,200,203,205]
**Turmeric**	Curcumin, an active ingredient of turmeric, has anti-inflammatory (H) and antioxidant properties (A). It has been shown to reduce IR in mice (A). This active ingredient significantly reduces ALT, AST, total cholesterol, LDL, fasting blood glucose, and insulin resistance (H).	[206,207,211,212,213]
**Garlic**	In animal studies, SAMC (active ingredient) has been linked to alleviating inflammation and insulin resistance (A). In human studies, garlic supplementation has been associated with improved levels of ALT, AST, total cholesterol, LDL cholesterol, TG, and fasting blood glucose (H).	[214,215,216,217]
**Basil, Lavender, Peppermint, Sage, Oregano, and Rosemary**	Ursolic acid (found in rosemary, peppermint, basil, lavender, and oregano) and carnosic acid (found in rosemary) have anti-inflammatory, antioxidant, and anti-apoptotic effects in animal studies (A).	[218,220]
**Ginger**	In animal studies, ginger has been associated with anti-lipogenic, anti-inflammatory, and antioxidant properties (A). In human trials, ginger supplementation has been shown to significantly improve ALT, AST, total cholesterol, LDL cholesterol, HDL cholesterol, triglycerides, insulin resistance, and hepatic steatosis (H).	[221,222,223]
**Gingko Biloba**	Ginkgo Biloba reduces oxidative stress and improves liver enzymes, hepatic steatosis, inflammation, and IR, as seen in animal studies (A).	[224,225,226]
**Ginseng**	Ginseng has been shown to improve liver enzyme function, thereby preventing hepatic inflammation, fibrosis, and steatosis in MASLD, as observed in animal studies (A).	[227,228,229]
**Licorice**	Chamomile and red clover may have hepatoprotective effects (A). Licorice is associated with improved IR and ALT levels (H).	[230,231]
**Plantago major**	Daily supplementation with 2 g of Plantago major seeds resulted in a substantial reduction in serum levels of ALT, TGs, and LDL, as well as alleviation of hepatic steatosis, compared to the placebo (H).	[233]
**Berberine**	Berberine (BBR) is an isoquinolone found in various medicinal plants; BBR improves intestinal barrier function and reduces inflammation caused by gut microbiota-derived LPS in metabolic diseases. This may improve glucose and lipid metabolism (H) (A). It has improved weight, HOMA-IR, AST, ALT, GGT, total cholesterol and LDL (H).	[234,235,236]
**OTHERS**		
**Probiotics**	Yogurt may improve IR, ALT, and hepatic fat in patients with MASLD. Probiotic/symbiotic use in MASLD may enhance liver steatosis, AST, ALT, endotoxins, and IR (H).	[241,242,243,244]
**Caffeine**	A Moderate amount of caffeine-containing coffee consumption (2–3 cups/day) decreased the severity of hepatic fibrosis and was associated with reduced risk of advanced liver fibrosis in MASLD (H).	[250,251,252]
**Green tea**	Daily supplementation with green tea extract may improve liver enzymes in patients with MASLD.	[255]
**Low-calorie Sweeteners**	The American Heart Association (AHA) and the American Diabetes Association (ADA) recommend reducing the consumption of sweeteners due to their adverse effects on body weight and cardiometabolic risk factors (H).	[260]
**Resveratrol**	Resveratrol in red wine has been shown to reduce oxidative stress, liver fat accumulation, and inflammation, as seen in animal models (A).	[261,262]
	Some randomized controlled trials examining the effect of daily resveratrol supplements indicated improvements in AST, ALT, and insulin resistance (H).	[263,264]
	Although it is associated with some improvement in inflammatory markers, it does not impact the overall management of MASLD (H).	[268]
**Choline**	Choline-deficient diets lead to intestinal dysbiosis and may be linked to MASH (H).	[270,271]
**Fish oil**	Daily fish oil supplementation (3 capsules each containing 0.315 g of omega-3 PUFAs) improved lipid profile, the function of liver enzymes, and steatosis (H).	[272]
**Co-enzyme Q10**	Co-enzyme Q10 daily supplementation (100 mg) is associated with reduced AST, GGT levels (H).	[276]
**Alcohol**	Heavy alcohol use (4 standard drinks/day or greater than 14 drinks/week in men or greater than three drinks/day or seven drinks/week in women as defined by NIAAA) is not recommended in patients with MASLD.	http://rethinkingdrinking.niaaa.nih.gov/How-much-is-too-much (accessed on 29 July 2025)
	Substantial evidence is not available to safely recommend light to moderate alcohol use in MASLD patients (H).	[277]
**Cannabinoids**	A notably lower prevalence of MASLD is reported among cannabis users, but more research is needed to confirm this effect (H).	[292,293]
**Tobacco**	Although a direct relationship between tobacco use and MASLD has not been found, it is considered a significant risk factor for HCC, CVD (H).	[297,298]

Abbreviations: ALT: alanine aminotransferase; AST: aspartate aminotransferase; BMI: body mass index; HbA1C: hemoglobin A1c; HDL: high-density lipoprotein; IHTG: intrahepatic triglycerides; IR: insulin resistance; LDL: low-density lipoprotein; MUFA: Monounsaturated fatty acid; MASLD: metabolic dysfunction-associated steatotic liver disease; MASH: non-alcoholic steatohepatitis; NIAAA: National Institute on Alcohol Abuse and Alcoholism; PUFA: polyunsaturated fatty acid; SAMC: S-ally mercaptocysteine; SFA: saturated fatty acid; TG: triglycerides; T2DM: type 2 diabetes mellitus; VLDL: very low-density lipoprotein.

**Table 3 ijms-26-09625-t003:** Exercise recommendations for MASLD patients by subgroup: Summary of exercise-related data in MASLD.

Patient Group	Recommended Modality	Target Dose (Duration/Intensity)	Ref.
**General Adults**	Aerobic, Resistance, or combined	150–300 min/week moderate aerobic (3–6 METs) OR 75–150 min vigorous (>6 METs); 2–3 RE sessions/week; Recommended activities include walking, cycling, jogging, swimming	[365,412]
**Sarcopenia/Muscle loss**	Resistance ± Aerobic	2–3 RE sessions/ week (50–75% 1RM)	[364]
**Older adults**	Walking/treadmill, low-moderate aerobic	~180 min/week (30 min/day × 6 days)	[413]
**Women**	Moderate aerobic, lifestyle activities	≥150 min/week	[414]
**Adolescents/Youth**	HIIT, aerobic sports	≥3 sessions/week	[415,416]
**Advanced fibrosis/comorbidities**	Aerobic ± RE (Supervised)	Individualized (≤moderate)	[417,418,419]
**“Weekend Warrior”**	Any	1–2 longer weekly sessions	[373]

Abbreviations: METs: Metabolic equivalents of task; RE: resistance exercise; 1RM: one-repetition maximum.

## Data Availability

No new data were created or analyzed in this study. Data sharing is not applicable to this article.

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
