# Peer review of "Diet and Lifestyle Interventions in Metabolic Dysfunction-Associated Fatty Liver Disease: A Comprehensive Review"

_ijms, 2025, doi:10.3390/ijms26199625_

Round 1

Reviewer 1 Report

Comments and Suggestions for Authors

The manuscript aim to review the role of lifestype interventions on MAFLD, the pathogenesis, the roles of dietary Modifications,macronutrients,micronutrients,minerals,herbal supplements,dietary pattern,physical activities and exercise,surgery etc. have been discussed. Some suggestions are given as below:

(1) Is the "17" at the end of the 4th paragraph in the Introduction an incorrectly formatted citation number? Please verify.   

(2) The discussion on semaglutide for pharmacological management of MASH is recommended to be included in the 5th paragraph of the Introduction.  

(3) Both "MASLD" and "MAFLD" are used throughout the manuscript; please maintain consistency by adopting one term.  

(4) The pathogenesis of MASLD lacks a clear summary and classification. While the authors appear to categorize it into genetic, metabolic, and environmental factors, several ambiguities remain: What are the specific environmental inducers? What are the precise inducers related to the spleen-liver axis? Mitochondrial dysfunction is a common mechanism. Factors like exercise, alcohol consumption, and gut microbiota are not properly classified. Additionally, the classification mentioned at the beginning of Section 3 ("Genetic, environmental, behavioral, and health-related factors are fundamental in the development of MAFLD") is inconsistent with that in Section 2. A precise classification of etiologies would effectively bridge the preceding context and subsequent analysis.  

(5) Figure 1, as presented for the corresponding content, does not align with the main text. Instead, it appears to illustrate the progression mechanisms of MAFLD-HCC rather than the pathogenesis of MASLD.  

(6)Some paragraphs are recommended to be merged to enhance content continuity, depth, and coherence. For instance, the last two paragraphs in Sections 3 or 4 could be combined.  

(7)Lifestyle interventions include weight loss,physical exercise, diet, and nutrients Since Section 3 mainly focuses on weight loss (rather than covering multiple lifestyle intervention types), the title of Section 3 is suggested to be revised for greater accuracy—ideally to align with its core content of weight loss.

(8) Please provide more details on the specific roles and mechanisms of minerals in MAFLD.  

(9) What is the role of circadian rhythm disturbance/staying up late in MAFLD? 

Comments on the Quality of English Language

The English could be improved to more concise to summarize the conclusion of cited papers.

Author Response

Changes to the manuscript

We sincerely thank the esteemed peer reviewers for dedicating their valuable time and expertise to evaluating our manuscript. Their insightful observations and constructive recommendations have substantially strengthened the quality and clarity of our work.

In addition to addressing their comments, we conducted a comprehensive internal critique, the main elements of which are outlined below. To facilitate tracking, all revisions have been clearly highlighted within the text. Furthermore, we have incorporated Key Summary Boxes to enhance readability and improve the manuscript’s accessibility for the reader.

Peer Reviewer # 1 Suggested changes:

  1. Is the "17" at the end of the 4th paragraph in the Introduction an incorrectly formatted citation number? Please verify.   

A: It was indeed an incorrectly formatted citation, which is now rectified

  1. The discussion on semaglutide for pharmacological management of MASH is recommended to be included in the 5th paragraph of the Introduction. 

A: The approval of Semaglutide as the second FDA-approved medication for MASLD has been added as recommended

  1. Both "MASLD" and "MAFLD" are used throughout the manuscript; please maintain consistency by adopting one term.  

A: Consistent use of a single term, “MASLD”, has been applied to the whole manuscript, unless indicated otherwise

  1. The pathogenesis of MASLD lacks a clear summary and classification. While the authors appear to categorize it into genetic, metabolic, and environmental factors, several ambiguities remain: What are the specific environmental inducers? What are the precise inducers related to the spleen-liver axis? Mitochondrial dysfunction is a common mechanism. Factors like exercise, alcohol consumption, and gut microbiota are not properly classified. Additionally, the classification mentioned at the beginning of Section 3 ("Genetic, environmental, behavioral, and health-related factors are fundamental in the development of MAFLD") is inconsistent with that in Section 2. A precise classification of etiologies would effectively bridge the preceding context and subsequent analysis.  

A: This crucial recommendation is now applied evenly in Sections 2 and 3

  1. Figure 1, as presented for the corresponding content, does not align with the main text. Instead, it appears to illustrate the progression mechanisms of MAFLD-HCC rather than the pathogenesis of MASLD.  

A: The rationale and scientific justification to incorporate the pathogenic mechanisms of hepatocellular carcinoma (HCC) as an integral aspect in the illustration of MASLD pathogenesis was to reflect the clinical continuum rather than artificially separating outcomes from underlying mechanisms. Robust recent evidence demonstrates that MASLD is now a leading etiology for HCC worldwide, with mechanistic links that extend seamlessly from hepatic steatosis, chronic inflammation, and fibrosis to malignant transformation

  1. Some paragraphs are recommended to be merged to enhance content continuity, depth, and coherence. For instance, the last two paragraphs in Sections 3 or 4 could be combined.  

A: These paragraphs were merged as per the recommendations

  1. Lifestyle interventions include weight loss, physical exercise, diet, and nutrients. Since Section 3 mainly focuses on weight loss (rather than covering multiple lifestyle intervention types), the title of Section 3 is suggested to be revised for greater accuracy—ideally to align with its core content of weight loss.

A: Section 3’s title and content have been revised; meanwhile, weight loss has been added as section 4. All subsequent sections have been down-numbered accordingly. This recommendation allowed better visualization and categorization of our manuscript.

  1. Please provide more details on the specific roles and mechanisms of minerals in MAFLD

A: Roles and mechanisms of minerals in MASLD have been expanded to include the recommendations in greater detail.

  1. What is the role of circadian rhythm disturbance/staying up late in MAFLD?

A: Section 8 has been added to address this recommendation

  1. The English could be improved to be more concise to summarize the conclusion of cited papers.

A: We have addressed this recommendation in a number of places, allowing a more concise, readable version of the manuscript.

Suggested changes by the internal team:

  • Abstract: Changes were made to the abstract to make it more comprehensive
  • Introduction: Apart from the recommended addition, this section was strengthened with quantitative epidemiologic data, reflecting the significance of diet and lifestyle in decreasing the clinical burden of MASLD at each stage.
  • Sections 3 & 4: Section 3’s content and title were revised as per recommendations by Peer-Reviewers. A separate section of “Role of Weight Loss” was added, whereby the topic was expanded in the context of Caloric restriction.
  • Section 5: Previously, it held the top of caloric restriction, which has been separated, as explained in the point above. Also, other sections related to diet have been switched as a subsection under this heading. All changes to the diet, whether recommended or not, have been included in this section.
  • Section 7: Section of exercise, including sarcopenia, types of exercise, etc, has been upgraded to include all the current and relevant information. Commonly known information has either been moved to the table or the Key Summary boxes, appropriately
  • Sections 8 & 9: New additions to the manuscript. As per our respected Peer-Reviewers and our team, to encapsulate the effects of Diet and Lifestyle interventions, these sections were added as per recent research.
  • Section 11: Future Directions, while maintaining the fidelity of ideas, has been upgraded to incorporate the recent direction the field of lifestyle intervention has taken

Reviewer 2 Report

Comments and Suggestions for Authors

The article submitted for review is very interesting. Although numerous review papers on fatty liver diseases have appeared in recent years, this work is perhaps the first such comprehensive study presenting all possible factors influencing the development of liver disease, and particularly its prevention. However, as a reviewer, I have a number of comments that must be addressed before the work can be published:

  • The title doesn't reflect the content. On page 7, section 3, line 2, it's clearly stated that the description concerns "diet and lifestyle" - and that's how the title should be, as a large part of the article is devoted to diet.
  • The paper presents a wealth of interesting information, but it is overly extensive. Sections 5-8 provide a comprehensive description of the role of macronutrients in MASLD, then summarized in Table 2. Most of the information about fats, sugars, and proteins has been widely known and described in the literature for many years. Therefore, there is no point in repeating this information. In my opinion, this section should be shortened to a more comprehensive table description.
  • Chapter 8 -Herbs. This chapter should also include critical opinions on some herbs. Recent research indicates adverse effects of herbs such as barberry.
  • In the work, the authors point out surgical treatment and its limitations. However, I miss a critical analysis of pharmacological agents, which have been very fashionable recently.

Author Response

Changes to the manuscript

We sincerely thank the esteemed peer reviewers for dedicating their valuable time and expertise to evaluating our manuscript. Their insightful observations and constructive recommendations have substantially strengthened the quality and clarity of our work.

In addition to addressing their comments, we conducted a comprehensive internal critique, the main elements of which are outlined below. To facilitate tracking, all revisions have been clearly highlighted within the text. Furthermore, we have incorporated Key Summary Boxes to enhance readability and improve the manuscript’s accessibility for the reader.

Peer Reviewer # 2 Suggested changes:

  1. The title doesn't reflect the content. On page 7, section 3, line 2, it's clearly stated that the description concerns "diet and lifestyle" - and that's how the title should be, as a large part of the article is devoted to diet.

A: This recommendation allowed us to expand on non-diet factors that play a role in MASLD. We also changed the title of the manuscript to reflect and encompass all the factors associated with MASLD

  1. The paper presents a wealth of interesting information, but it is overly extensive. Sections 5-8 provide a comprehensive description of the role of macronutrients in MASLD, then summarized in Table 2. Most of the information about fats, sugars, and proteins has been widely known and described in the literature for many years. Therefore, there is no need to repeat this information. In my opinion, this section should be shortened to a more comprehensive table description.

A: Our team agreed with this recommendation, and we have condensed our sections related to readily available knowledge in the section on fats, sugars, and proteins. All pertinent points were added to the table

  1. Chapter 8 -Herbs. This chapter should also include critical opinions on some herbs. Recent research indicates the adverse effects of herbs such as barberry.

      A: This recommendation has been addressed in the Key summary of herbs

  1. In the work, the authors point out surgical treatment and its limitations. However, I miss a critical analysis of pharmacological agents, which have been very fashionable recently.

A: While we also enjoy quantitative synthesis, this recommendation allowed us to rethink the surgical section in MASLD, and we concluded that it is beyond the scope of the article. Hence, it was removed. Although other sections, which are pertinent to our topic, were added, e.g., Role of Circadian rhythm, role of behavioral and psychosocial determinants

Suggested changes by the internal team:

  • Abstract: Changes were made to the abstract to make it more comprehensive
  • Introduction: Apart from the recommended addition, this section was strengthened with quantitative epidemiologic data, reflecting the significance of diet and lifestyle in decreasing the clinical burden of MASLD at each stage.
  • Sections 3 & 4: Section 3’s content and title were revised as per recommendations by Peer-Reviewers. A separate section of “Role of Weight Loss” was added, whereby the topic was expanded in the context of Caloric restriction.
  • Section 5: Previously, it held the top of caloric restriction, which has been separated, as explained in the point above. Also, other sections related to diet have been switched as a subsection under this heading. All changes to the diet, whether recommended or not, have been included in this section.
  • Section 7: Section of exercise, including sarcopenia, types of exercise, etc, has been upgraded to include all the current and relevant information. Commonly known information has either been moved to the table or the Key Summary boxes, appropriately
  • Sections 8 & 9: New additions to the manuscript. As per our respected Peer-Reviewers and our team, to encapsulate the effects of Diet and Lifestyle interventions, these sections were added as per recent research.
  • Section 11: Future Directions, while maintaining the fidelity of ideas, has been upgraded to incorporate the recent direction the field of lifestyle intervention has taken

Round 2

Reviewer 2 Report

Comments and Suggestions for Authors

Authors included all reviewer comments. The paper is very interesting and well written. No further comments.